# Normalized Difference Vegetation Index Maps of Pure Pixels over China's mainland for Estimation of Fractional Vegetation Cover

Tian Zhao[1,2], Wanjuan Song[3], Xihan Mu[1,2*], Yun Xie[4,5], Yuanyuan Wang[6], Hangqi Ren[1,2], Donghui Xie[1,2], Guangjian Yan[1,2]

[1]State Key Laboratory of Remote Sensing and Digital Earth, Faculty of Geographical Science, Beijing Normal University, Beijing 100875, China
[2]Beijing Engineering Research Center for Global Land Remote Sensing Products, Faculty of Geographical Science, Beijing Normal University, Beijing 100875, China
[3]State Key Laboratory of Remote Sensing and Digital Earth, Aerospace Information Research Institute, Chinese Academy of
Sciences, Beijing 100101, China
[4]State Key Laboratory of Earth Surface Processes and Resource Ecology, Faculty of Geographical Science, Beijing Normal University, Beijing 100875, China
[5]College of Arts and Sciences, Beijing Normal University, Zhuhai 519087, China
[6]Key Laboratory of Radiometric Calibration and Validation for Environmental Satellites/Key Laboratory of Space Weather,
National Satellite Meteorological Center (National Center for Space Weather), Innovation Center for FengYun Meteorological Satellite (FYSIC), Beijing 100875, China

*Correspondence to*: Xihan Mu (muxihan@bnu.edu.cn)

**Abstract.** Fractional Vegetation Cover (FVC) is an important vegetation structure factor for applications in agriculture,
forestry, ecology, etc. Due to its simplicity, the normalized difference vegetation index (NDVI)-based mixture model is widely used to estimate FVC from remotely sensed data. However, the accuracy and efficiency of FVC estimation require the precise calculation of two key parameters: the NDVI of fully covered vegetation and bare soil. Despite their importance, these two endmember NDVI values have not yet been produced as large-scale maps. Traditional statistical methods for obtaining endmember NDVI from satellite datasets highly rely on the assumption that a certain amount of pure pixels of vegetation and
soil must be present, which is often invalid for many areas. This study generated 30 m resolution maps of endmember NDVI across China's mainland using the MultiVI algorithm, incorporating multi-angle remote sensing data. The quality and accuracy of the endmember NDVI maps were evaluated using various validation data, including statistically obtained pure NDVI, soil spectra from a soil library, and field-measured FVC. The NDVI values for bare soil derived from the MultiVI algorithm were consistent with those obtained from the soil spectral library. Additionally, the FVC estimated using the MultiVI-derived
endmember NDVI and the VI-based mixture model exhibited reasonable accuracy compared to the field measurements. The root mean square deviation (RMSD) values for MultiVI FVC were below 0.13 in the Heihe, Hebei, and Three Gorges Reservoir regions of China. These regions include typical arid and humid zones, facilitating the evaluation of the algorithm's performance under diverse climatic conditions. Furthermore, the MultiVI FVC outperformed those calculated using the traditional statistical methods. The endmember NDVI maps provide a convenient and reliable source of key parameters for the accurate and rapid

estimation of FVC at large scales. The 30 m pure NDVI maps of 2014 are publicly available at https://zenodo.org/records/14060222 (Zhao et al., 2024).

## 1 Introduction

Fractional vegetation cover (FVC) quantitatively characterizes the horizontal density of photosynthetically active vegetation (Gutman and Ignatov, 1997). It is typically defined as the planar proportion of green vegetation to the total surface extent
(Deardorff, 1978). The FVC is an essential parameter in climate and hydrologic models as it represents the spatial contribution of vegetation (Hirano et al., 2004; Gutman and Ignatov, 1998; Eriksson et al., 2006; Mölders and Olson, 2004). Accurate and high-resolution FVC products are in high demand for various studies, including climate change analysis, soil erosion assessment, land disturbance evaluation, and crop growth monitoring (Xie et al., 2011; Naqvi et al., 2013; Gan et al., 2014; Li et al., 2014; Zhang et al., 2013; Fernández-Guisuraga et al., 2021).
Remote sensing can rapidly and repeatedly observe the land surface, making estimating FVC on regional or global scales feasible. Over recent decades, tremendous efforts have been made to derive high-quality FVC from remotely sensed imagery. The published approaches for retrieving FVC can generally be summarized as follows: (i) the vegetation index (VI)-based mixture model (Gutman and Ignatov, 1998; Zeng et al., 2000; Wu et al., 2014; Mu et al., 2021; Zhao et al., 2023); (ii) spectral mixture analysis (García-Haro et al., 2005; Dimiceli et al., 2011; Guan et al., 2012); (iii) machine learning (Baret et al., 2007;
Baret et al., 2013; Jia et al., 2015); and (iv) physical model (Xiao et al., 2016).
The linear mixture model is the most commonly used spectral unmixing method. It is generally utilized in surface elements evaluation such as vegetation classification, surface disturbance mapping, and evapotranspiration estimation (Li et al., 2018a; Lu et al., 2003; Cochrane and Souza Jr, 1998). When considering only two endmembers (green vegetation and bare soil), linear mixture modelling can be employed to calculate the relative abundance of live vegetation from the mixed VI, known as the
VI-based mixture model. It assumes that the VI for a particular pixel originates from a linearly weighted sum of green vegetation and bare soil, with their respective areal proportions as weighting coefficients (Gitelson et al., 2002). The mixed VI ($V$) of the pixel is linearly decomposed by the two endmembers, $i.e.$, the VI of the fully vegetated ($Vv$) and bare soil pixel ($Vs$), to obtain the areal proportion of green vegetation as FVC:

$$FVC = \frac{V - V_s}{V_v - V_s} \ ,  \tag{1}$$

Despite being the most commonly used method for deriving FVC (Gao et al., 2020), the VI-based mixture model still requires enhancements in both accuracy and efficiency. A major limitation is the challenge of obtaining accurate values for $Vv$ and $Vs$ on a large scale. The "greenness" VIs that exhibit a strong linear relationship with FVC can be used in the VI-based mixture model, such as the enhanced vegetation index (EVI) and the normalized difference vegetation index (NDVI) (Song et al., 2022a). Among them, the NDVI is the primary VI used to derive FVC due to its strong correlation with vegetation structural
parameters (Gutman and Ignatov, 1998). The two endmember NDVI values (hereafter referred to as pure NDVI values) are

often assigned *a priori*, and the main methods for extracting these values, along with other pure VIs are summarized in Table 1.

**Table 1 Brief summary of the primary methods for determining the two pure VI values (*Vv* and *Vs* with NDVI as the default VI).**

| Methods | *Vv* | *Vs* | Reference |
|---|---|---|---|
| Independent field spectral measurements of pure vegetation and bare soil pixels | 0.71 | 0.16 | (Wang and Qi, 2008) |
| | | *The values are of Modified Soil-Adjusted Vegetation Index (MSAVI). | |
| Visual interpretation to identify pure pixels from high-resolution remotely sensed images | 0.78 | 0.03 | (Jiao et al., 2014) |
| The endmember extraction algorithm, *e.g.*, the pure pixel index (PPI) method | 0.941 | 0.068 | (Jia et al., 2017) |
| The cumulative maximum and minimum, or cumulative percentages of NDVI values derived from remotely sensed datasets within a specific area or time series | **Vv determination for different land types:** The 90th percentile of the annual maximum NDVI for different land types (shrubland, barren, sparsely vegetated), the 75th percentile (other) **Vv values for each land type:** 0.49 (grassland) 0.60 (open shrubland) 0.68 (mixed forest) 0.70 (broadleaf and deciduous forests) | **Vs determination for all land types:** The 5th percentile of the annual maximum NDVI for barren and sparsely vegetated land areas **Vs values for all land types:** 0.05 | (Zeng et al., 2000) |
| | **Vv determination for different land types:** The 98th percentile of the monthly maximum NDVI over 5 years for different land types **Vv values for each land type:** 0.752 (crop, grass, desert, shrub) 0.816 (mixed woodland, forest) 0.824 (broadleaf deciduous) | **Vs determination for all land types:** The 2nd percentile of the monthly maximum NDVI over 5 years for desert and semi-desert **Vs values for all land types:** 0.048 | (Oleson et al., 2000) |

| | *Vv* determination for different biomes: | *Vs* determination for different biomes: | (Matsui et al., 2005) |
|---|---|---|---|
| | The 97th percentile of the historical maximum over 20 years for different biomes | The 3rd percentile of the historical minimum over 20 years for different biomes | |
| | *Vv* values for each biome: | *Vs* values for each biome: | |
| | 0.52 (arid) | 0.03 (arid) | |
| | 0.72 (seasonal) | 0.04 (seasonal) | |
| | 0.67 (evergreen) | 0.05 (evergreen) | |

Traditional methods for determining pure NDVI values have significant limitations. Collecting *Vv* and *Vs* from ground truth data or high-resolution remotely sensed imagery by extracting pure pixel values is time-consuming and often limited by data availability. These methods become ineffective when data are unavailable or there are no pure pixels in the usable dataset. The statistical method infers the two pure NDVI values from the data within a specific area or time series. It typically assumes that pixels with the lowest NDVI values represent bare soil, while those with the highest values represent pure vegetation (Gao et al., 2020). However, in arid and semiarid regions with few fully vegetated pixels, or evergreen forests with limited bare soil pixels, the statistically extracted endmember values can be significantly inaccurate (Song et al., 2017).

Additionally, statistical methods typically assign a single value of *Vv* or *Vs* for a specific region or land type. However, endmember values can vary significantly from pixel to pixel due to differences in species composition, vegetation health, moisture levels, and other factors (Jensen, 2000). In many studies, a fixed value of *Vs* is adopted for various soil types, as shown in Table 1, which overlooks the spatial variations in soil moisture, texture, mineralogy, organic matter, and other characteristics (Yang and Yang, 2006; Zeng et al., 2000). The NDVI values of soil samples exhibit considerable variation, ranging from 0 to 0.4, with a mean value of 0.2, which is significantly higher than the commonly used *Vs* value (of approximately 0.05) (Montandon and Small, 2008). Notably, the accuracy of FVC estimation is highly sensitive to variations of *Vs*, particularly in sparse-vegetated areas (Asrar et al., 1984; Montandon and Small, 2008). Underestimating *Vs* may lead to an overestimation of FVC, with errors reaching up to 20% in grassland and shrubland regions (Montandon and Small, 2008; Ding et al., 2016). It has been demonstrated that locally derived pure NDVI values provide higher accuracy for FVC estimation, compared to using a fixed global *Vs* value (Donohue et al., 2014; Montandon and Small, 2008). Therefore, pixel-wise maps of endmember NDVI are essential for effectively addressing the spatial variability of plant and soil reflectance. However, such products are currently unavailable.

Recent studies have proposed an alternative method that uses multi-angle datasets from the Moderate Resolution Imaging Spectroradiometer (MODIS) to derive pixel-wise *Vv* and *Vs* (MultiVI algorithm) (Mu et al., 2018; Song et al., 2022a). The discrepancies in directional NDVI values observed from multiple viewing geometries imply vegetation structural information and soil characteristics (Chen et al., 2005; Diner et al., 1999; Deering, 1999; Verrelst et al., 2008). The MultiVI algorithm utilizes the variations from two large viewing angles to establish equations for simultaneously retrieving *Vv* and *Vs*, without

assuming invariant endmember values within a scene or biome. Feasibility analysis indicated its potential to estimate high-quality, high-resolution FVC products (Song et al., 2022b). The *Vv* and *Vs* derived from the MultiVI algorithm have been used to generate 30-m/15-day FVC products, which are reported to possess satisfactory accuracy (Zhao et al., 2023). However, the accuracy of the *Vv* and *Vs* maps still requires evaluation, and strategies to optimize the MultiVI algorithm for large-scale *Vv* and *Vs* mapping are essential. Providing high-quality pure NDVI values for the VI-based mixture model will significantly improve the accuracy and efficiency of FVC estimation.

This study aims to generate and validate 30 m resolution pixel-wise *Vv* and *Vs* maps across China's mainland. These datasets can be applied to accurately calculate FVC at various resolutions on regional or national scales. The *Vv* and *Vs* maps were derived from MODIS reflectance data using the MultiVI algorithm. Subsequently, the 500 m MODIS *Vv* and *Vs* were downscaled to 30 m resolution based on land cover types (hereafter referred to as MultiVI *Vv* and *Vs*). Traditional statistical methods were employed to extract *Vv* and *Vs* from Landsat data (hereafter referred to as statistical *Vv* and *Vs*). The two sets of pure NDVI values were then compared and analysed. The generated *Vs* maps were validated using soil NDVI values calculated from a soil spectral library. Finally, the FVC values derived from the MultiVI algorithm and the statistical method were validated against field-measured FVC obtained from various experimental sites across China's mainland.

## 2 Datasets

### 2.1 Satellite data for *Vv* and *Vs* calculation

#### 2.1.1 Terra/Aqua MODIS BRDF products

The MODIS Bidirectional Reflectance Distribution Function (BRDF) / Albedo Model Parameters product (MCD43A1) and its corresponding quality assessment product (MCD43A2) (https://www.earthdata.nasa.gov/data/catalog/lpcloud-mcd43a1-061, last accessed 17 June 2025) are the primary datasets used to derive pure NDVI values using the MultiVI algorithm. These datasets are produced daily using atmospherically corrected, cloud-cleared input data from the Terra and Aqua satellites over 16 days at 500 m resolution. The BRDF characterizes surface anisotropic scattering as a function of illumination and viewing angles. The MCD43A1 product contains three sets of model weighting parameters, *i.e.*, the RossThick kernel (volume-scattering kernel), LiSparseR kernel (geometric-optical kernel), and isotropic kernel parameters. These parameters can be used with the semiempirical linear kernel-driven model, known as the semiempirical RossThick-LiSparse Reciprocal (RTLSR), to calculate surface reflectance (SR) for any required viewing and illumination directions (Roujean et al., 1992; Schaaf et al., 2002). All MCD43A1 data obtained in 2014 over China's mainland were used to reconstruct the ground surface reflectance of red and near-infrared (NIR) bands at viewing zenith angles (VZAs) of 55° and 60° (see Section 3.1.1 for further details on the selection of angular configuration). These reflectance values were subsequently used to generate directional NDVI for the MultiVI algorithm. The quality assessment data from MCD43A2 were applied to exclude clouds, snow, water, and low-quality pixels.

### 2.1.2 GlobeLand30 datasets

A global land cover dataset with a 30 m resolution, known as GlobeLand30, was used to downscale the 500 m resolution pure NDVI values to 30 m. The GlobeLand30 products are available globally in three versions: 2000, 2010, and 2020, with the 2020 version being adopted for this study (https://doi.org/10.12041/geodata.140236667788805.ver1.db, last accessed 17 June 2025). These products were developed and updated using cloudless or minimally cloudy multispectral images from Landsat, HJ-1, and GF-1 (Chen et al., 2014; Chen et al., 2015). Validation based on over 230,000 samples indicated a total accuracy of 85.72% for the 2020 version of GlobeLand30, with a Kappa coefficient of 0.82. The GlobeLand30 products define bare land as having an FVC of lower than 10%, which is stricter than the criteria used by other land cover products. This criterion helps minimize the misclassification of sparse shrubland or grassland as bare land (Liu et al., 2021). The GlobeLand30 product categorizes land cover into ten classes: six vegetation classes (*i.e.*, cultivated land, forest, grassland, shrubland, wetland, and tundra) and four non-vegetation classes (*i.e.*, artificial surfaces, bare land, water bodies, and perennial snow and ice) (Jun et al., 2014). The classes of wetlands, water bodies, and perennial snow and ice were grouped during the downscaling of $Vv$ and $Vs$, as their pure NDVI values are generally similar (below 0).

## 2.2 Validation data

### 2.2.1 Statistical $Vv$ and $Vs$

The Landsat 8 data were used to obtain $Vv$ and $Vs$ using statistical methods on the Google Earth Engine (GEE) platform. These statistical $Vv$ and $Vs$ maps were subsequently compared to the pure NDVI values derived from the MultiVI algorithm. The Landsat 8 Collection 2 SR products with a resolution of 30 m provided atmospherically corrected SR data (https://www.usgs.gov/landsat-missions/landsat-collection-2-level-2-science-products, last accessed 17 June 2025). The time-series Landsat 8 SR images from 2010 to 2020 were analysed to derive statistical $Vv$ and $Vs$. Pixels identified as cloud, cloud shadow, water, and snow in the Landsat 8 images were excluded using the corresponding quality assessment data.

### 2.2.2 Soil NDVI from the soil spectral library

The soil spectra measured at the Soil and Plant Spectral Diagnostic Laboratory of the World Agroforestry Center (ICRAF) were used to calculate soil NDVI values (https://files.isric.org/public/other/, last accessed 17 June 2025), which were subsequently compared with the retrieved $Vs$ values. The soil spectral library that includes soil samples collected from 58 countries was developed by ICRAF in collaboration with the International Soil Reference and Information Centre (ISRIC). This soil spectral library provides laboratory-measured soil spectra, along with attribute data such as geographical coordinates, horizon, and physical and chemical properties. Approximately 20 g of air-dried soil samples were passed through a 2 mm sieve, and placed into 7.4 cm diameter Duran glass Petri dishes for measurement (Garrity and Bindraban, 2006). Spectral measurements were conducted using a FieldSpec FR spectroradiometer (Analytical Spectral Devices, Boulder, CO) at wavelengths ranging from 0.35 to 2.5 µm, with 1 nm intervals (Garrity and Bindraban, 2006).

Among the soil samples in the ICRAF soil spectral library, 247 were collected in China's mainland (Figure 1). The retrieved *Vs* values were constrained to be no lower than 0, consistent with the findings of most studies (Montandon and Small, 2008; Ding et al., 2016). Consequently, those samples with NDVI values below 0 were excluded, resulting in a total of 228 samples available for validation. These samples were categorized into eight soil types (Table 2) based on their collection locations and

the soil type map of China's mainland (https://www.resdc.cn/data.aspx?DATAID=145, last accessed 17 June 2025). Additionally, Figure 1 illustrates the ecological and geographical zones of China's mainland, highlighting the humid and arid areas according to moisture conditions (https://www.resdc.cn/data.aspx?DATAID=125, last accessed 17 June 2025).

The NDVI values of the soil samples in the ICRAF soil spectral library range from 0 to 0.273, with the majority concentrated around 0.1. The mean NDVI value for all soil samples in China's mainland is 0.077, with a standard deviation of 0.049. Desert

soils exhibit the lowest NDVI values, with a maximum of 0.039. This soil type is mainly found in the arid regions of northwestern China (Figure 1). In contrast, Alfisols have the highest NDVI values, reaching a maximum of 0.273. These soils are primarily distributed in the humid regions of northeastern and southwestern China (Figure 1).

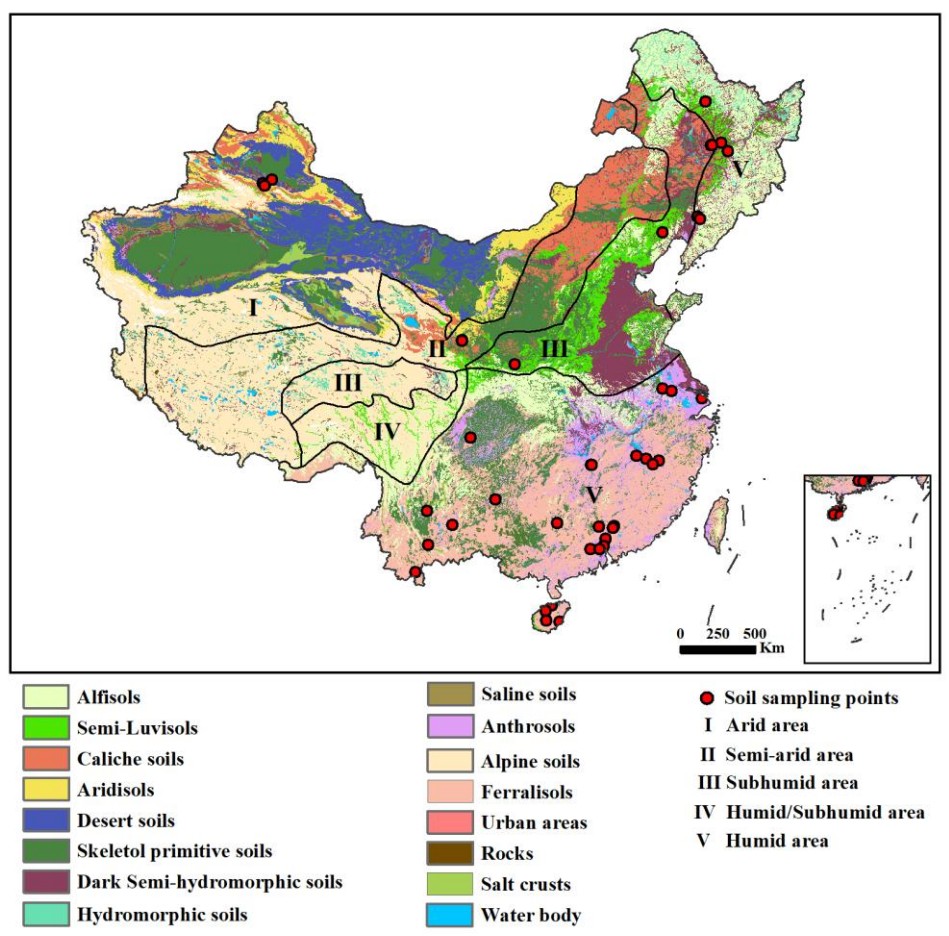

**Figure 1: The spatial distribution of soil types in China's mainland. The red circle represents the locations of soil samples in the**
**ICRAF soil spectral library. The ecological and geographical zones of China's mainland are numbered using Roman numerals.**

**Table 2 NDVI of the soil samples collected from the soil spectral library.**

| Soil type | Number of samples | Mean | Maximum | Minimum | Standard deviation |
|---|---|---|---|---|---|
| Alfisols | 10 | 0.158 | 0.273 | 0.055 | 0.070 |
| Semi-Luvisols | 22 | 0.109 | 0.206 | 0.051 | 0.039 |
| Caliche soils | 12 | 0.108 | 0.206 | 0.044 | 0.061 |
| Desert soils | 10 | 0.028 | 0.039 | 0.012 | 0.011 |
| Skeletol primitive soils | 28 | 0.081 | 0.163 | 0.016 | 0.047 |
| Dark Semi-hydromorphic soils | 9 | 0.142 | 0.204 | 0.057 | 0.049 |
| Anthrosols | 41 | 0.058 | 0.110 | 0.001 | 0.031 |
| Ferralisols | 96 | 0.064 | 0.180 | 0.001 | 0.037 |

### 2.2.3 Field-measured FVC

The field-measured FVC data were collected from three sites that represent typical climatic conditions in China: (i) the Hebei watershed in humid northeastern China, (ii) the Heihe River Basin in arid northwestern China, and (iii) the Three Gorges Reservoir Area in humid southwestern China. All field measurements were conducted over a complete vegetation growth cycle, with data collected approximately every 15 days. Figure 2 shows the locations of these sites.

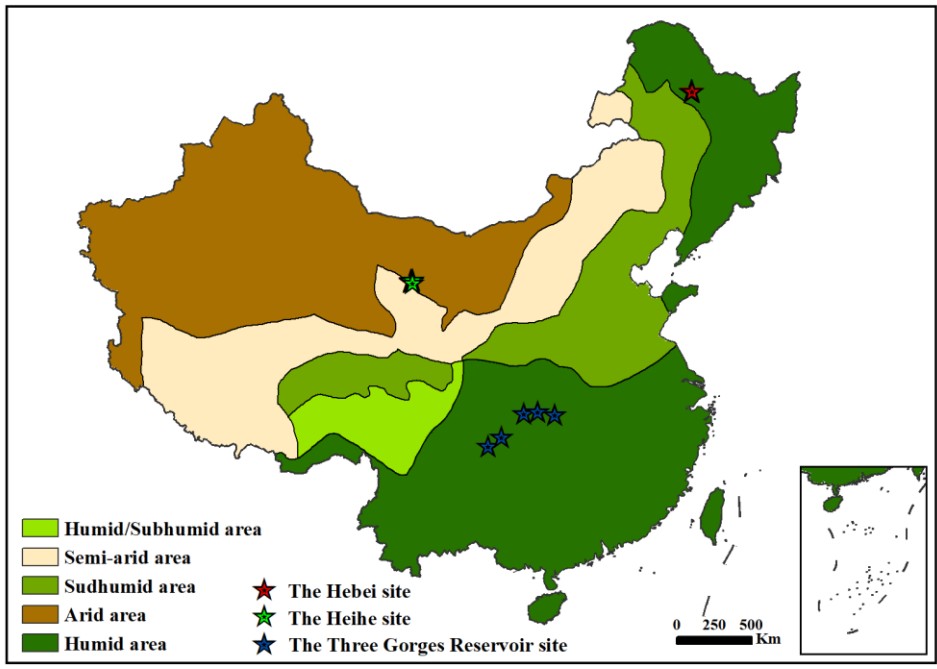

**Figure 2: The ecological and geographical zones of China's mainland. The red, green, and blue pentagrams symbolize the Hebei site, the Heihe site and the Three Gorges Reservoir site, respectively.**

The Hebei watershed is located in a typical black soil region of China and is mainly covered by crops. Eight sampling plots were distributed in relatively flat and uniform areas, comprising five forest plots, two grassland plots, and one cropland plot. The field measurements were conducted between 15 April and 30 October, 2022. For the cropland and grassland, an unmanned aerial vehicle (UAV) was used to capture FVC images, achieving a resolution higher than 1.5 cm and covering a plot size of 100 m × 100 m. The UAV-derived FVC values were used in the subsequent validation analysis. For the forests, a digital camera was used to acquire FVC, with a plot size of 30 m. The camera was mounted on a long pole at a height of 1.5 to 2 meters above the ground and took vertical photographs at regular intervals along two diagonal lines within each sampling plot (Mu et al., 2013; Li et al., 2012). Images were captured from top to bottom and bottom to top at each step to document the coverage of understory vegetation ($f_{up}$) and overstory canopy ($f_{down}$), respectively. The FVC was calculated as the weighted sum of the $f_{up}$ and $f_{down}$ using the following Eq. (2):

$$FVC = f_{up} + \left(1 - f_{up}\right) \cdot f_{down} , \tag{2}$$

The study area in the Heihe River Basin comprises 72% cropland, 24% residential land, and 4% woodland, indicating significant surface heterogeneity (Mu et al., 2015). The sampling plots, each measuring 10 m × 10 m, were exclusively located in cropland areas within the Heihe Watershed Allied Telemetry Experimental Research (HiWATER) sites, where corn was the predominant crop. The FVC measurements were conducted using a digital camera from 15 May to 14 September, 2012, encompassing the entire growing season.

The Three Gorges Reservoir Area is situated in the mid-upper reaches of the Yangtze River. Five representative small river basins were selected for vegetation monitoring within this region. Seasonal trajectories of vegetation cover were recorded using UAVs every two weeks from 15 August, 2021, to 1 August, 2022. Each small river basin contained a sampling plot of approximately 100 m × 100 m, distributed in a relatively flat and homogeneous area. The primary vegetation types included orchards, forests, and shrubland. The UAV images had a resolution of higher than 1.5 cm.

A Half-Gaussian Fitting algorithm (HAGFVC) was used to calculate FVC from UAV-acquired images, resulting in a minimal mean bias error (MBE) and root mean square error (RMSE) of less than 0.04 (Li et al., 2018b). The digital images captured by hand-held cameras were processed using a shadow-resistant algorithm (SHAR-LABFVC) to extract FVC, achieving an RMSE of approximately 0.025 (Song et al., 2015).

## 3 Methods

### 3.1 MultiVI algorithm for retrieving the pure NDVI maps

#### 3.1.1 Theory

The MultiVI algorithm uses multi-angle remotely sensed observations to retrieve $V_v$ and $V_s$. It defines the directional vegetation cover, $F(\theta)$, which represents the FVC at the VZA $\theta$. A nonlinear coefficient, $k$, is introduced in the VI-based

mixture model. This coefficient accommodates the potential nonlinear relationship between FVC and NDVI, as shown in Eq. (3) (Xiao and Moody, 2005; Jiapaer et al., 2011; Choudhury et al., 1994; Mu et al., 2024):

$$F(\theta) = \left(\frac{V(\theta) - V_s}{V_v - V_s}\right)^k, \tag{3}$$

where $V(\theta)$ is the NDVI observed at VZA $\theta$. The directional gap fraction model can be expressed as Eq. (4) (Nilson, 1971):

$$P(\theta) = e^{-G(\theta)\cdot\Omega(\theta)\cdot LAI/\cos\theta}, \tag{4}$$

Here, $P(\theta)$ denotes the directional gap fraction, $G(\theta)$ is the mean projection of unit foliage area (Goel and Strebel, 1984), $\Omega(\theta)$ is the clumping index, and LAI represents the leaf area index. The directional FVC and gap fraction exhibit a complementary relationship, such that the sum of $F(\theta)$ and $P(\theta)$ equals 1. Therefore, Eqs. (3) and (4) can be combined as follows:

$$1 - \left(\frac{V(\theta) - V_s}{V_v - V_s}\right)^k = e^{-G(\theta)\cdot\Omega(\theta)\cdot LAI/\cos\theta}, \tag{5}$$

The $G(\theta)$ in Eq. (5) is often assumed to be constant at large VZAs around 57.5°, despite variations in leaf angle distributions (Leblanc et al., 1999; He et al., 2011; Song et al., 2017; Mu et al., 2018; Weiss et al., 2004; Roujean et al., 1992). Furthermore, the variation of $G(\theta)\cdot\Omega(\theta)$ is significantly smaller than the angular variation of cosθ at large VZAs (Mu et al., 2018). Since the LAI is also independent of VZA, $G(\theta)\cdot\Omega(\theta)\cdot LAI$ can be assumed to be invariant at large VZAs around 57.5°. The angular effects of $Vv$ and $Vs$ are negligible (Escadafal and Huete, 1991; Mu et al., 2018). By using pairs of observations at large VZAs around 57.5° and eliminating angle-invariant parameters, Eq. (5) can be reorganized as:

$$\left(1 - \left(\frac{V(\theta_i) - V_s}{V_v - V_s}\right)^k\right)^{\cos\theta_i} = \left(1 - \left(\frac{V(\theta_j) - V_s}{V_v - V_s}\right)^k\right)^{\cos\theta_j}, \tag{6}$$

where the subscripts "$i$" and "$j$" represent pairs of large VZAs around 57.5°. The combination of 55° and 60° in the forward viewing directions was identified as the optimal angular configuration. This selection is attributed to its minimal influence on $G(\theta)$ and the high quality of angular remote sensing observations (Mu et al., 2018). These angles were used to formulate equations for estimating $Vv$ and $Vs$:

$$\left(1 - \left(\frac{V(55°) - V_s}{V_v - V_s}\right)^k\right)^{\cos 55°} = \left(1 - \left(\frac{V(60°) - V_s}{V_v - V_s}\right)^k\right)^{\cos 60°}, \tag{7}$$

The unknown parameters $Vv$, $Vs$, and $k$ for a given pixel can be derived using at least three pairs of angular observations at VZAs of 55° and 60° to solve Eq. (7).

### 3.1.2 Implementation

The observed NDVI used to solve Eq. (7) should exhibit distinct variations to ensure stable results (Mu et al., 2018). This study used observations from different periods to construct appropriate equations for each pixel. The $Vv$ and $Vs$ were independently retrieved to enhance accuracy by applying high and low NDVI values, respectively. The minimum and maximum NDVI values for each pixel in 2014 were used as statistical boundaries. This approach ensured that the derived $Vs$

values remained below the annual minimum, while the derived *Vv* values exceeded the annual maximum. Furthermore,

empirical boundaries for *Vs* ([0.01, 0.3]) and *Vv* ([0.6, 1.0]) were applied to constrain the retrieval, preventing the upper limit of *Vs* from exceeding 0.3 and the lower limit of *Vv* from falling below 0.6 (Montandon and Small, 2008). This flexible approach integrates statistical boundaries for *Vv* and *Vs*, while allowing for reasonable intra-variability within each land cover type. Figure 3 illustrates the steps for using the MultiVI algorithm to derive 500 m pure NDVI values from the MODIS BRDF products:

(1) Daily directional NDVI values at VZA of 55° and 60° were calculated for each pixel throughout the year using the semi-empirical RTLSR model;

(2) NDVI pairs at VZA of 55° and 60° were ranked in ascending order based on the values at 55° VZA for the entire year;

(3) The annual NDVI value sequence was divided into two groups based on the 10th percentile. The NDVI values below the 10th percentile were used to retrieve *Vs* using Eq. (7), while the remaining 90% were used to retrieve *Vv*;

(4) The 25th, 50th, 75th, and 100th percentiles of the NDVI pairs in each group were selected to construct inversion equations (Eq. 7) for each pixel. The unknown parameters *Vv*, *Vs*, and *k* were numerically solved using the least squares method;

(5) For a small number of invalid pixels due to limited observations, gap filling was performed based on the MODIS land cover data (MCD12Q1). The mean values of *Vv* and *Vs* corresponding to the same land cover type were used to fill these gaps.

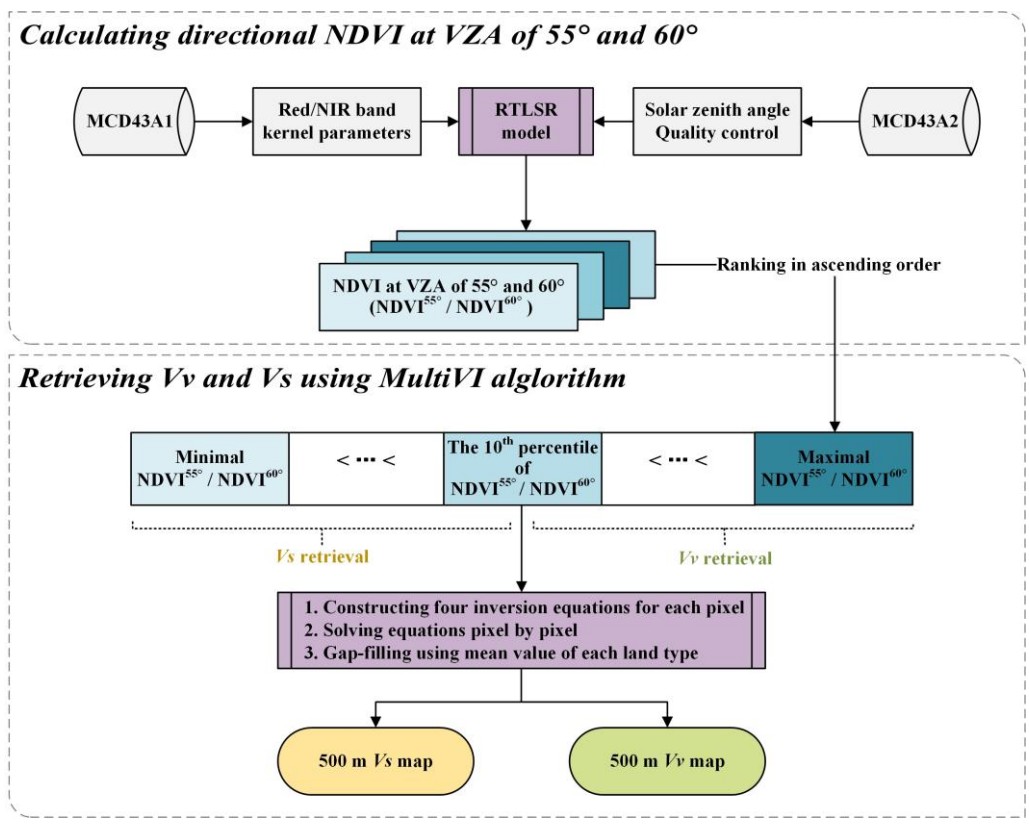

**Figure 3: The scheme of the MultiVI algorithm to derive 500 m pure NDVI values from MODIS BRDF products.**

## 3.2 NDVI unmixing for downscaling the 500 m *Vv* and *Vs*

The 500 m *Vv* and *Vs* were downscaled to a 30 m resolution using NDVI unmixing to facilitate fine-scale FVC estimation. It was assumed that the same land cover type within each MODIS pixel was assigned the same *Vv* and *Vs* values. The 500 m *Vv* and *Vs* were considered as linear combinations of the 30 m *Vv* and *Vs* values within that MODIS pixel, with weights determined by the areal proportions of land cover types.

The downscaling process utilized seven land cover types from the GlobeLand30 product, specifically four vegetation classes (cultivated land, forest, grassland, shrubland, and tundra), a grouped water surface category (wetland, water body, and permanent snow and ice), bare land, and artificial surfaces. A $3 \times 3$ sliding window with a step size of one MODIS pixel was employed to construct an overdetermined system of unmixing equations for the MODIS pixel at $(x, y)$, as shown in Eq. (8) for downscaling *Vv*:

$$\begin{cases} V_{v,\text{modis},x-1,y-1} = \sum_{i=1}^{n} P_{i,x-1,y-1} V_{v,i,x,y} \\ V_{v,\text{modis},x-1,y} = \sum_{i=1}^{n} P_{i,x-1,y} V_{v,i,x,y} \\ V_{v,\text{modis},x-1,y+1} = \sum_{i=1}^{m} P_{i,x-1,y+1} V_{v,i,x,y} \\ V_{v,\text{modis},x,y-1} = \sum_{i=1}^{n} P_{i,x,y-1} V_{v,i,x,y} \\ V_{v,\text{modis},x,y} = \sum_{i=1}^{n} P_{i,x,y} V_{v,i,x,y} \\ V_{v,\text{modis},x,y+1} = \sum_{i=1}^{n} P_{i,x,y+1} V_{v,i,x,y} \\ V_{v,\text{modis},x+1,y-1} = \sum_{i=1}^{n} P_{i,x+1,y-1} V_{v,i,x,y} \\ V_{v,\text{modis},x+1,y} = \sum_{i=1}^{n} P_{i,x+1,y} V_{v,i,x,y} \\ V_{v,\text{modis},x+1,y+1} = \sum_{i=1}^{n} P_{i,x+1,y+1} V_{v,i,x,y} \end{cases} , \tag{8}$$

where $V_{v,\text{modis},x,y}$ represents the *Vv* of a target MODIS pixel, *i.e.*, the central MODIS pixel of the sliding window. Each equation corresponds to a neighbouring MODIS pixel in the $3 \times 3$ window. $P_{i,x,y}$ signifies the areal proportion of the *i*th land cover type within this MODIS pixel, indicating its area-weighted contribution. $V_{v,i,x,y}$ denotes the *Vv* for the *i*th land cover type, which is assumed to be constant across the sliding window for each land cover type *i*, and is an unknown to be estimated.

By solving these equations, the *Vv* value was inferred for each land cover type. The derived values of $V_{v,i,x,y}$ were then assigned as the *Vv* for all the 30 m pixels of land cover type *i* within the MODIS pixel at $(x, y)$. The same procedure was applied to obtain $V_{s,i,x,y}$. Figure 4 illustrates the method for downscaling 500 m pure NDVI values to 30 m resolution.

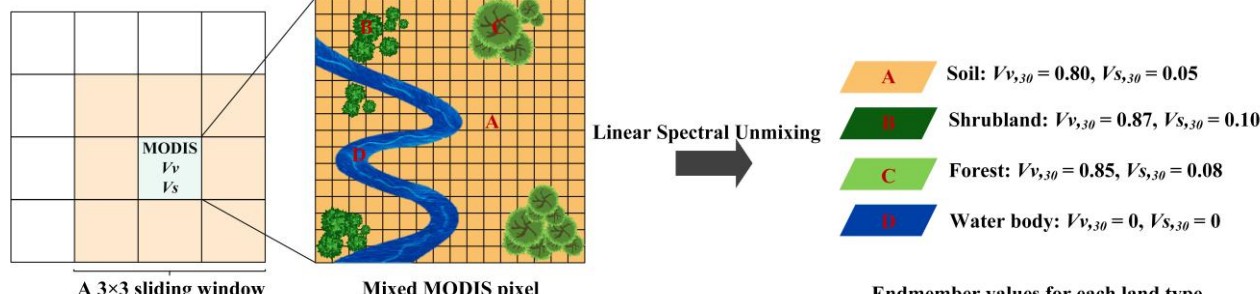

**Figure 4: The schematic diagram for downscaling the 500 m MultiVI *Vv* and *Vs* to 30 m resolution based on the GlobeLand30 product.**

### 3.3 Assessment and validation

#### 3.3.1 Comparison with statistical methods

Statistical methods were used to generate 30 m statistical $Vv$ and $Vs$ in 2014 across China's mainland using Landsat 8 data, which were then compared with the MultiVI $Vv$ and $Vs$. The statistical method utilized Landsat 8 data from 2010 to 2020, processed on the Google Earth Engine (GEE) platform. The pixel-wise maximum and minimum NDVI values from 2010 to 2020 were set as the $Vv$ and $Vs$, respectively. The statistical $Vv$ and $Vs$ were compared with the MultiVI $Vv$ and $Vs$ to assess their differences in spatial patterns and magnitudes.

#### 3.3.2 Comparison with soil NDVI from the soil spectral library

The soil spectra obtained from the soil spectral library were convolved to the red and NIR bands using the spectral response functions of MODIS. The convolved soil spectra were then used to calculate the soil NDVI for comparison with the retrieved $Vs$. The MultiVI $Vs$ and statistical $Vs$ were averaged for each soil type to facilitate comparison with the soil NDVI. Additionally, the uncertainties in the estimated FVC caused by intra-class variability in $Vs$ were assessed for each soil type.

#### 3.3.3 Assessment with field-measured FVC

The field-measured FVC at the Hebei, Heihe, and Three Gorges Reservoir sites were used to assess the estimated FVC derived from the MultiVI $Vv/Vs$ and that from the statistical $Vv/Vs$. The Landsat 8 NDVI time series were smoothed using a harmonic model (Zhao et al., 2023). The NDVI was generated at half-monthly intervals to coincide with the observation times of the field measurements. Subsequently, the NDVI was converted to FVC using the VI-based mixture model and the retrieved $Vv$ and $Vs$.

The plot sizes for ground measurements ranged from 10 m to 100 m, and various spatial matching strategies were implemented to enhance the consistency between ground observations and satellite-derived FVC. For the cropland and grassland plots at the Heihe site, as well as all plots at the Three Gorges Reservoir site, UAV measurements with a plot size of 100 m × 100 m were utilized. To minimize potential edge effects and ensure spatial correspondence, UAV-derived FVC was processed to orthophotos, calculated over the central 90 m × 90 m area and compared with the average FVC from the co-located 3 × 3 block of 30 m satellite pixels. For forest plots measuring 30 m × 30 m at the Heihe site, satellite FVC was averaged over a surrounding 3 × 3 window to account for potential geolocation uncertainty and scale mismatch, following the methodology of Weiss et al. (Weiss et al., 2007). Given the high spatial homogeneity of the Heihe site, direct pixel-to-plot comparisons were also performed by matching the ground-measured FVC from the 10 m × 10 m plots with the corresponding 30 m satellite pixels. The correlation coefficient ($R^2$) and the root mean square deviation (RMSD) were used to assess the relationship and differences between the field-measured FVC and the estimated FVC, respectively.

**4 Results**

**4.1 Maps of the MultiVI *Vv* and *Vs***

Figure 5 shows the *Vv* and *Vs* maps generated using the MultiVI algorithm (Figures 5a and 5b) and the statistical method (Figures 5c and 5d). The MultiVI *Vv* and *Vs* maps demonstrate smooth distributions, whereas the statistical *Vv* and *Vs* maps

exhibit noticeable stripes along the borders of the Landsat 8 imagery tiles. The spatial patterns of pure NDVI values derived from the MultiVI algorithm and the statistical method show similar trends, predominantly influenced by moisture conditions. Specifically, the humid and sub-humid areas of southeastern China are characterized by elevated pure NDVI values, while the arid and semi-arid regions of northwestern China are dominated by lower values.

The statistical *Vv* are generally lower than the MultiVI *Vv* in most areas, particularly in semi-arid and arid regions. In

northwestern China, which is primarily covered by grasslands, bare lands, and deserts, the statistical method yields NDVI values of less than 0.3 for pure vegetation pixels (Figure 3b). These values are significantly lower than the *Vv* values reported in most studies (Table 1).

Table 3 presents the MultiVI *Vv* and *Vs* values across various land types. The mean *Vv* values range from 0.817 to 0.909, while the mean *Vs* values range from 0.074 to 0.287. Both MultiVI *Vv* and *Vs* exhibit consistent patterns across different land types,

indicating that vegetation with higher *Vv* values tends to also exhibit higher *Vs* values. The shrublands show the lowest values, whereas evergreen needleleaf forests exhibit the highest values. The forests generally have higher values than other land types, with evergreen forests surpassing broadleaf forests. In contrast, grasslands and shrublands have lower values compared to other land types. The standard deviation values for *Vs* are greater than those for *Vv*.

The spatial patterns of the MultiVI *Vs* exhibit distinct variations across different soil types and demonstrate a closer alignment

with the actual soil distribution in China's mainland (Figure 1) when compared to the statistical *Vs*. The MultiVI *Vs* are generally lower than the statistical *Vs*, especially in the densely vegetated areas of southeastern China. In these humid regions, the statistical *Vs* values exceed 0.4, where evergreen vegetation predominates and shows few bare lands (Figure 3d). These values, which exceed 0.4, are notably higher than the generally accepted soil NDVI values (Table 1).

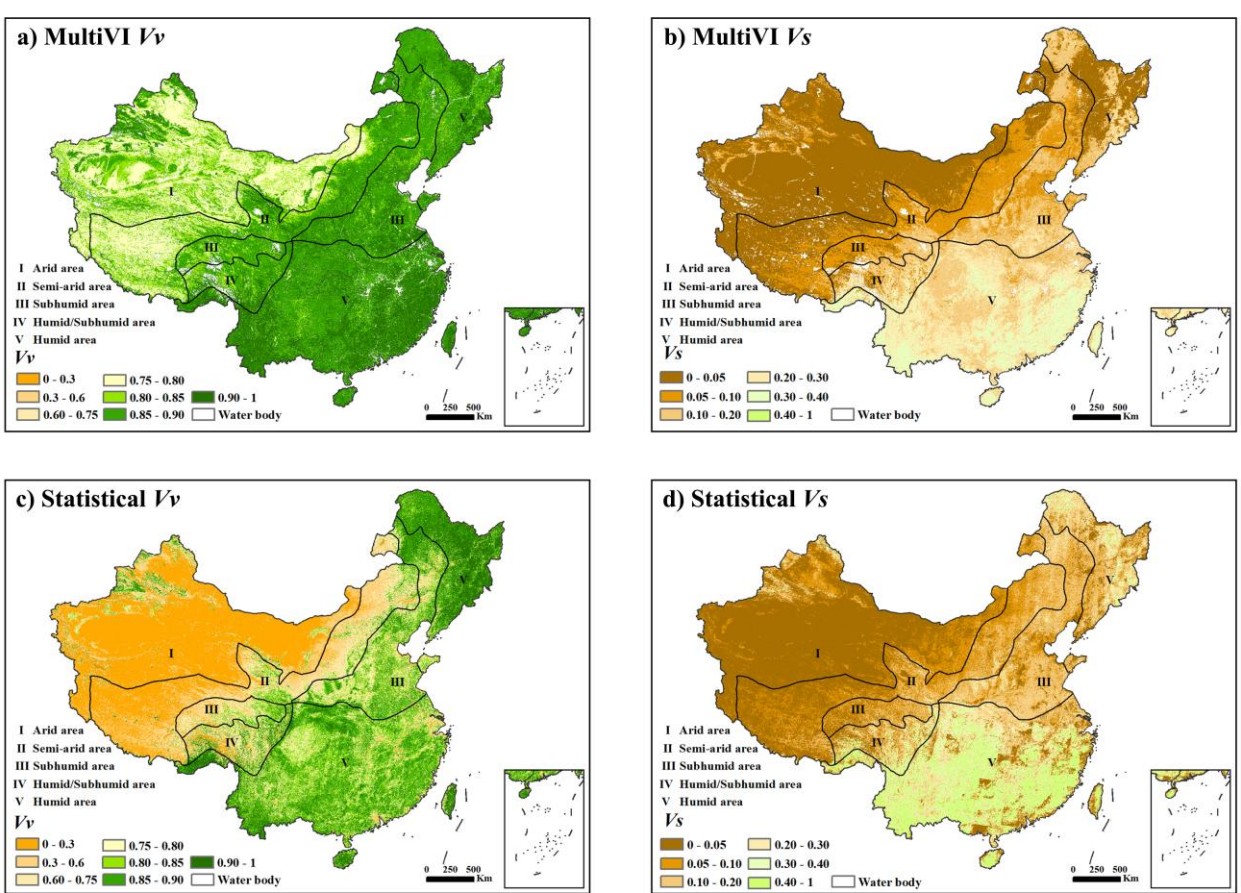

**Figure 5: The spatial distributions of *Vv* and *Vs* generated using the MultiVI algorithm and the statistical method, respectively.**

**Table 3 MultiVI *Vv* and *Vs* for different land types.**

| Land type | mean *Vv* | standard deviation of *Vv* | mean *Vs* | standard deviation of *Vs* |
|---|---|---|---|---|
| Grassland | 0.857 | 0.046 | 0.074 | 0.056 |
| Shrublands | 0.817 | 0.047 | 0.068 | 0.044 |
| Broadleaf croplands | 0.887 | 0.031 | 0.123 | 0.079 |
| Savananas | 0.893 | 0.032 | 0.219 | 0.084 |
| Evergreen broadleaf forests | 0.907 | 0.029 | 0.314 | 0.060 |
| Deciduous broadleaf forests | 0.894 | 0.027 | 0.174 | 0.086 |
| Evergreen needleleaf forests | 0.909 | 0.032 | 0.287 | 0.073 |
| Deciduous needleleaf forests | 0.875 | 0.022 | 0.156 | 0.070 |

Figure 6 shows the differences between the MultiVI $Vv/Vs$ and the statistical $Vv/Vs$. The largest discrepancies are observed in regions lacking pure pixels, *i.e.*, the arid areas where pure vegetation pixels are absent for $Vv$ estimation, and the humid areas where bare soil pixels are lacking for $Vs$ estimation.

In the sparse grasslands of northwestern China, the MultiVI $Vv$ are significantly higher than the statistical $Vv$, with a bias exceeding 0.3 (Figure 6a). In most humid or subhumid areas of southeastern China, the difference between the two sets of $Vv$ values is generally within ±0.1. For relatively sparse vegetation, such as grasslands and croplands, the MultiVI $Vv$ are slightly higher than the statistical $Vv$. However, in forested areas, the MultiVI $Vv$ are slightly lower than the statistical $Vv$.

In the densely vegetated forests of southeastern China, the MultiVI $Vs$ are markedly lower than the statistical $Vs$, with a bias of less than -0.3 (Figure 6b). In arid regions, the MultiVI $Vs$ values are slightly lower than the statistical $Vs$ values in sparse grasslands and bare lands, but higher in oases, with biases of approximately -0.1 and +0.1, respectively.

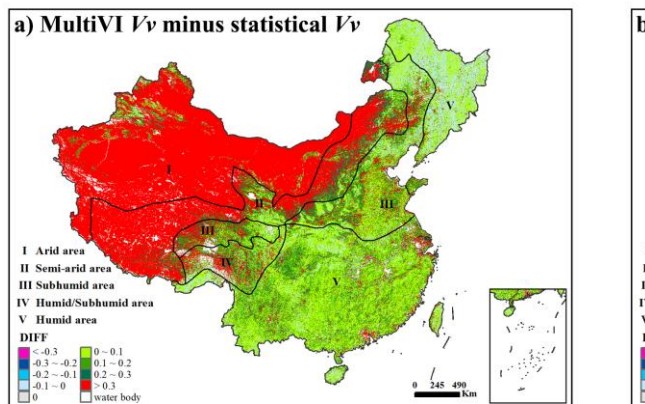 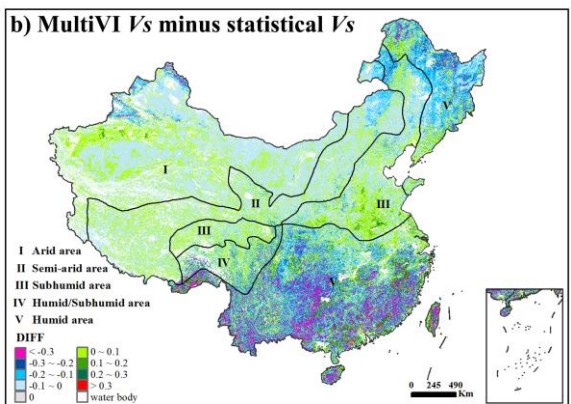

**Figure 6: The difference maps between the MultiVI NDVI and statistical NDVI for pure vegetation and bare soil.**

**4.2 Evaluation with soil NDVI from the soil spectral library**

Figure 7 shows the MultiVI $Vs$ and statistical $Vs$ in comparison to the soil NDVI derived from the ICRAF soil library. For Alfisols and Semi-Luvisols, the median of the MultiVI $Vs$ closely aligns with the median NDVI of the corresponding soil samples. In contrast, the median statistical $Vs$ tend to slightly overestimate the soil NDVI. For Desert soils and Skeletol primitive soils, the median of the statistical $Vs$ is closer to the soil NDVI, while the MultiVI $Vs$ exhibit a slight underestimation. Both the median values of the MultiVI and the statistical $Vs$ are lower than the median NDVI for Dark Semi-hydromorphic soils, with a bias of approximately 0.1. Soil samples of Anthrosols and Ferralisols are primarily distributed in the humid, densely vegetated regions of southeastern China, which results in relatively high NDVI values. For these soil types, the MultiVI $Vs$ show an overestimation when compared to the median NDVI values, with biases of approximately 0.15 for Anthrosols and 0.2 for Ferralisols. For Ferralisols, this overestimation is more pronounced in the statistically derived $Vs$, with biases exceeding 0.3.

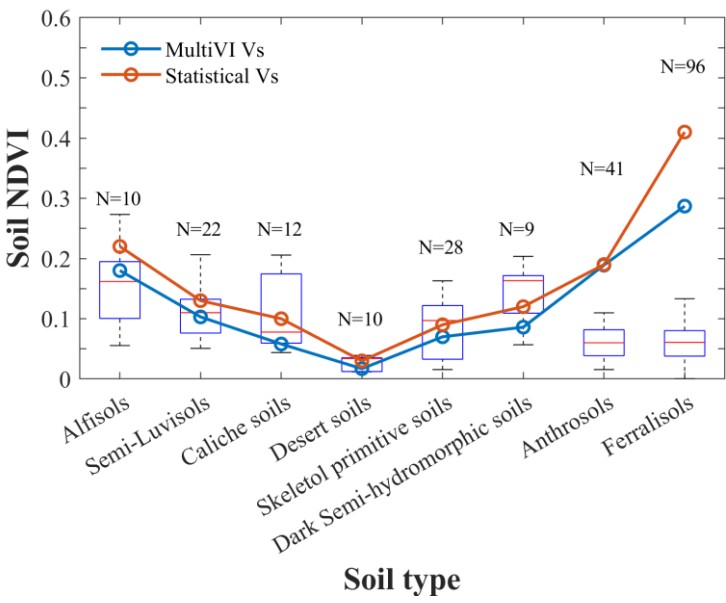

**Figure 7: The boxplot of the soil NDVI from the ICRAF soil library for each soil type. Each boxplot features a central red line representing the median. The N above the box indicates the number of sampling plots for each soil type. The lower and upper edges of the box denote the 25<sup>th</sup> and 75<sup>th</sup> percentiles, respectively. The whiskers are extended to the most extreme data points, excluding outliers. The blue and red lines denote the median values of the MultiVI *Vs* and statistical *Vs*, respectively.**

Figure 8 illustrates the bias between the MultiVI FVC and the statistical FVC compared to the reference FVC estimated using soil NDVI from the ICRAF soil library. The FVC bias represents the difference between the MultiVI FVC (Figure 8a) or statistical FVC (Figure 8b) and the reference FVC. The reference FVC was estimated using the ICRAF soil NDVI in combination with either MultiVI *Vv* (Figure 8a) or statistical *Vv* (Figure 8b).

For the MultiVI FVC, the median bias is within ±0.05, and the mean absolute values consistently remain below 0.1 across all soil types. The overestimation of the MultiVI *Vs* yields a slight underestimation of FVC for Anthrosols and Ferralisols. The overestimations of 0.15 and 0.2 in MultiVI *Vs* leads to underestimations of approximately 0.04 and 0.08 in FVC for Anthrosols and Ferralisols, respectively, both of which are located in densely vegetated areas. Conversely, the slight underestimation of MultiVI *Vs* relative to soil NDVI results in overestimations of 0.07 and 0.08 in FVC for Caliche soils and Dark Semi-hydromorphic soils, respectively. The statistical method performs better than the MultiVI algorithm for the Alfisols, Caliche soils, and Dark Semi-hydromorphic soils. The underestimation of statistical FVC exceeds -0.1 for Skeletol primitive soils, Anthrosols, and Ferralisols.

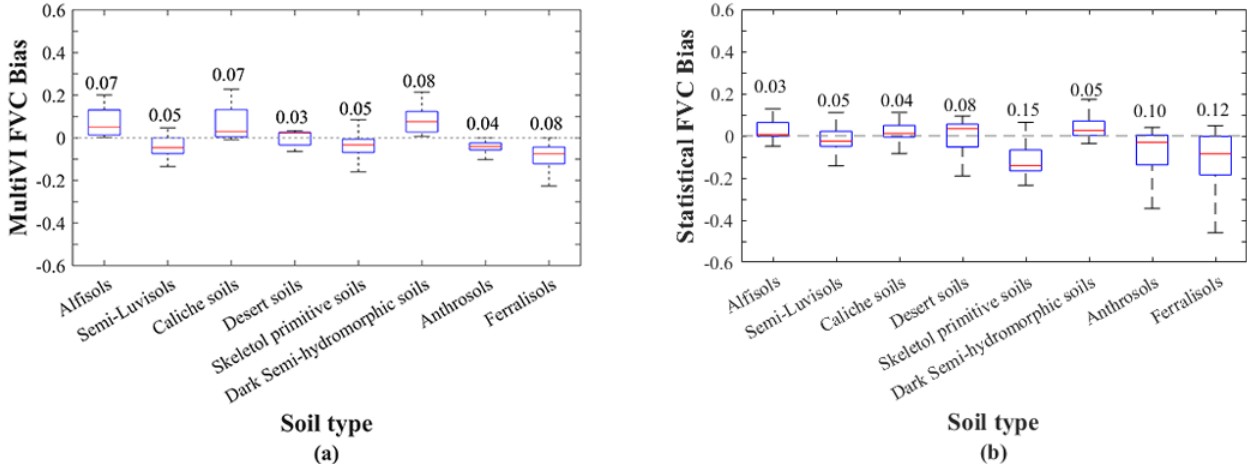

**Figure 8: The bias in FVC estimation across different soil types using different *Vs* values. The FVC bias represents the difference between the estimated FVC using (a) MultiVI *Vs* or (b) statistical *Vs* and the reference FVC derived from soil NDVI in the ICRAF soil library. The number above each box indicates the mean absolute bias of FVC for each soil type. The lower and upper edges of the box denote the 25th and 75th percentiles, respectively. The whiskers are extended to the most extreme data points, excluding outliers.**

## 4.3 Accuracy of FVC estimation

Figure 9 depicts scatterplots that compare field-measured FVC with FVC derived from the MultiVI *Vv/Vs* and the statistical *Vv/Vs* across three sites: the Hebei site (Figures 9a and 9b), the Heihe site (Figures 9c and 9d), and the Three Gorges Reservoir site (Figures 9e and 9f).

The MultiVI FVC demonstrates superior accuracy relative to the statistical FVC at the Hebei site, exhibiting a lower RMSD of 0.0938 for the MultiVI FVC, in contrast to 0.1491 for the statistical FVC. Additionally, the MultiVI FVC has a higher $R^2$ of 0.8121, compared to 0.7156 for the statistical FVC. Both the MultiVI FVC and the statistical FVC show saturation effects for high FVC values at the Hebei site during the summer season (Figures 9a and 9b). However, the MultiVI FVC aligns more closely with the 1:1 line, particularly during the non-growing seasons, specifically in April, September, and October.

The MultiVI FVC exhibits a slightly lower RMSD of 0.1285 compared to 0.1694 for the statistical FVC at the Heihe site. However, it has a lower $R^2$ of 0.7712 versus 0.8033 for the statistical FVC. The majority of the data points depicted in Figures 9c and 9d are situated above the 1:1 line, indicating an overestimation in both the MultiVI FVC and the statistical FVC.

At the Three Gorges Reservoir site, the MultiVI FVC also demonstrates superior accuracy, as indicated by an RMSD of 0.1022, in contrast to 0.1801 for the statistical FVC. Additionally, the correlation between the MultiVI FVC and field-measured FVC is significantly higher ($R^2 = 0.8162$) than that of the statistical FVC ($R^2 = 0.6572$). As illustrated in Figure 9f, a part of the statistical FVC data points are saturated at high values and drop to zero at low values, indicating that the statistical method suffers from an underestimation of *Vv* and an overestimation of *Vs* in humid areas.

Figure 9 also reveals seasonal patterns in the errors of FVC derived from the VI-based mixture model. At the Hebei site, larger deviations from the 1:1 line are observed in April and October, indicating greater uncertainty during the early growth and senescence stages. At the Heihe site, the FVC shows the largest error in June, particularly when 0.2<FVC<0.5. In the Three Gorges Reservoir area, the most significant FVC errors occur during winter months (January to February) and summer months (July to September). The seasonal patterns in the errors of the statistical FVC are generally more pronounced than those of the MultiVI FVC.

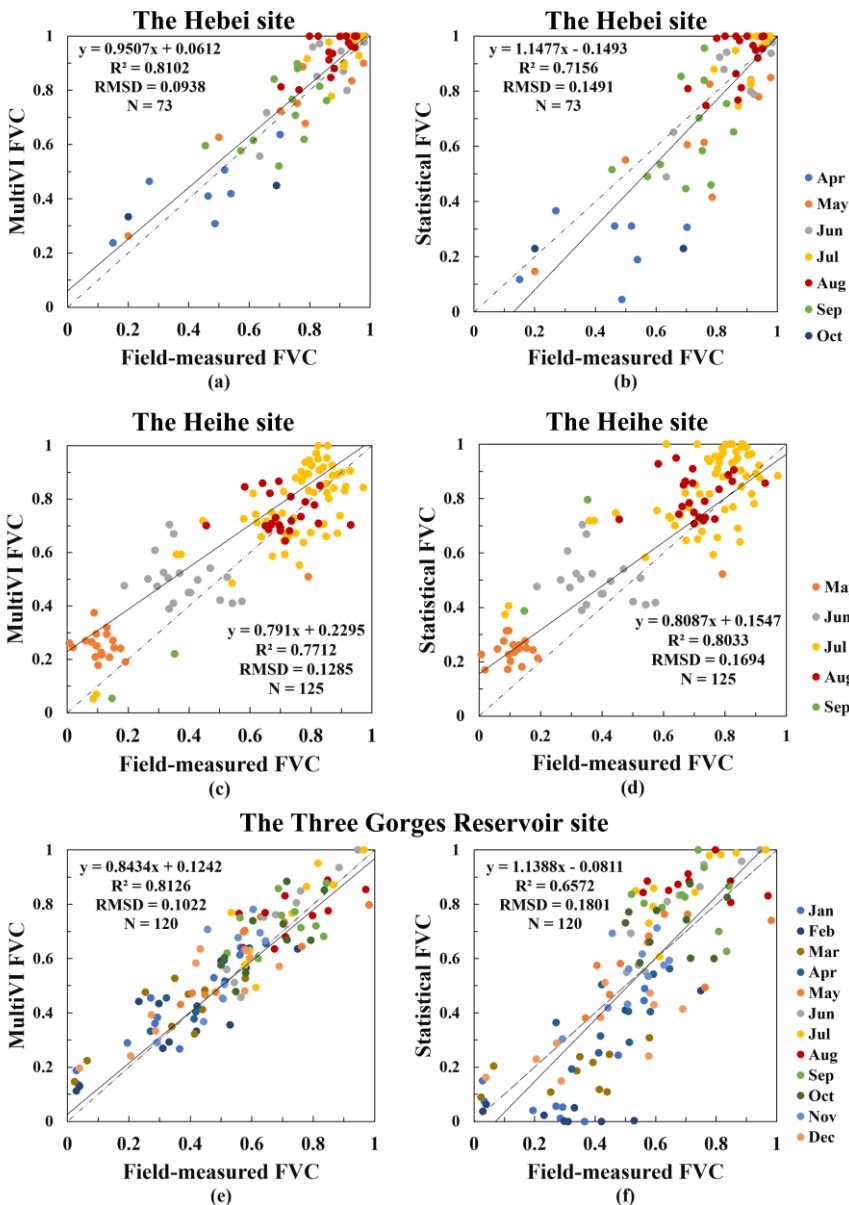

**Figure 9: Scatterplots of the MultiVI FVC and the statistical FVC versus the field-measured FVC. N is the number of samples.**

## 5 Discussion

The VI-based mixture model is widely used to estimate FVC due to its ease of implementation. The model's simplicity is primarily attributed to the preselection of two critical parameters: the NDVI of bare soil ($Vs$) and that of pure vegetation ($Vv$). These parameters are essential for the model's performance and have a significant impact on its accuracy. The traditional statistical method for obtaining $Vv$ and $Vs$ assumes the existence of pure pixels, either spatially or temporally. This method employed extreme values from regional or temporal datasets to represent $Vv$ and $Vs$. However, this approach has two significant

limitations: (1) pure pixels may be absent in certain ecosystems, such as evergreen forests, where bare soil pixels are lacking, or in bare lands, where pure vegetation pixels are absent; (2) the $Vv$ and $Vs$ can vary from pixel to pixel, yet the traditional statistical method often assigns a single value for an entire region or land type.

The global soil spectral library was used to calibrate $Vs$ and to account for its spatial variability (Montandon and Small, 2008). Although this method improves the accuracy of FVC estimation compared to using a single $Vs$ value across large regions, it is

415 limited in that it only considers soil variability within the spectral library and fails to address the spectral differences among individual pixels. An alternative approach is to use each pixel's historical lowest NDVI values as $Vs$ to ensure pixel-wise variability. In practice, $Vs$ is influenced not only by mineral soil properties but also by non-photosynthetic vegetation (NPV), biological residues, and surface litter. A previous study reported that the NDVI of NPV endmembers is generally higher than that of bare soil, with differences reaching up to 0.4 in some cases (Tian et al., 2021). This suggests that $Vs$ may vary even

within the same soil type. Land cover classification can partially account for such heterogeneity, and land cover data were utilized as a practical proxy to disaggregate 500 m $Vs$ to a 30 m resolution in this study.

Figure 7 further illustrates that the retrieved $Vs$ values deviate from the soil NDVI in humid regions. The most significant discrepancy is observed in the soil types of Anthrosols and Ferralisols, which are prevalent in the humid areas of southern China. This discrepancy is likely attributed to their relatively high NDVI values, and the presence of surface litter and

425 biological residues. However, the overestimation of MultiVI $Vs$ in these areas has a limited impact on the FVC estimation (Figure 8a) , with the resulting FVC bias generally less than 0.1. Previous studies have also indicated that the influence of $Vs$ on FVC estimation is more pronounced in areas with low NDVI values, such as grasslands and croplands / natural vegetation areas, compared to regions with high NDVI values (Ding et al., 2016).

In this study, statistical maps of $Vv$ and $Vs$ were generated using a common statistical criterion (Section 3.3.1). Empirical

NDVI for fully vegetated pixels is typically reported to exceed 0.5 in most studies (Gao et al., 2020; Zeng et al., 2000; Montandon and Small, 2008), whereas the statistical $Vv$ are observed to be below 0.3 in arid regions of China's mainland (Figure 5a). Except for these extreme areas, the statistical method demonstrates comparable accuracy to the MultiVI algorithm (Figures 7, 8, and 9c, 9d). The simple statistical methods face challenges in identifying appropriate NDVI values when pure pixels are lacking. In contrast, several studies have adopted more sophisticated statistical approaches to produce accurate FVC

estimates across various ecosystems (Donohue and Renzullo, 2025; Donohue et al., 2018). These enhanced statistical methods take into account the inherent characteristics of vegetation in different ecosystems, employing separate thresholds to determine

endmembers in arid and vegetated areas, respectively. The use of spatially adaptive thresholding facilitates the acquisition of reasonable endmembers and achieves high FVC accuracy, with a RMSE of approximately 0.1 (Donohue and Renzullo, 2025).

The errors in the FVC estimated from the VI-based mixture model vary across seasons and are more pronounced during periods of low vegetation cover. Figure 9 shows that FVC errors are generally larger in early spring and winter, particularly for low FVC values. This seasonal pattern can be explained by the sensitivity of FVC estimation to NDVI values in the 0.2–0.4 range (Montandon and Small, 2008). Within this interval, small errors in $Vs$ can lead to systematic overestimations of FVC, especially over grasslands and shrublands. During peak growing seasons, when NDVI values exceed 0.7, the model estimates become more stable and less sensitive to endmember NDVI values. Furthermore, the saturation of FVC estimates, resulting from an underestimation of $Vv$ and an overestimation of $Vs$, typically influences winter and summer periods (Figure 9f).

The MultiVI algorithm, which uses multi-angle data to retrieve $Vv$ and $Vs$, has demonstrated effectiveness in estimating FVC and has been applied to generate FVC time series on a national scale (Song et al., 2022a; Zhao et al., 2023). This algorithm retrieves angle-invariant NDVI values for endmembers using observations from two large VZAs. In this study, the retrieval strategy of the MultiVI algorithm was optimized through the construction of well-posed equations based on NDVI time series. Additionally, the statistical values of $Vv$ and $Vs$ were incorporated as boundary constraints during the equation-solving process. The downscaling procedure integrates a 30 m land cover product, which introduces finer spatial detail to describe the sub-pixel heterogeneity within MODIS pixels. Our downscaling approach applies the assumption that the 30 m pixels of the same land cover type share constant $Vv$ and $Vs$ values within a localized 3×3 window of 500 m MODIS pixels. This assumption may introduce uncertainty when significant soil type variation exists within the 3×3 MODIS window (1.5 km × 1.5 km). Despite the simplifications involved, the comparison with the 500 m results shows that the downscaled 30 m $Vs$ values achieve comparable accuracy (Song et al., 2022b). The downscaling process introduces minimal changes to the endmember values (Song et al., 2022b). Considering the increasing demand for high spatial and temporal resolution applications, providing 30 m endmember products is of practical significance.

As a critical vegetation parameter, FVC is frequently demanded in various studies, serving as an input for models or as fundamental data for ecological analyses. The VI-based mixture is one of the simplest methods for converting remotely sensed images into FVC products, provided that the pure NDVI values are obtained beforehand. However, the two essential parameters in the VI-based model, $Vv$ and $Vs$, currently lack standardized and reliable data sources. The newly generated 30 m $Vv$ and $Vs$ maps address this gap. Users can calculate accurate FVC values flexibly using the MultiVI $Vv$ and $Vs$ with their NDVI to meet specific requirements.

## 6 Conclusion

This study demonstrates the feasibility of generating pixel-wise NDVI for fully vegetated area ($Vv$) and bare soil ($Vs$) for the VI-based mixture model. In this study, 30 m resolution maps of pure NDVI values for the year 2014 were produced for China's mainland using multi-angle remotely sensed data from MODIS and land cover datasets from Globeland30. The assessment

and validation of the *Vv* and *Vs* maps were conducted from three aspects: 1) comparing the MultiVI *Vv* and *Vs* maps with those generated using the statistical method; 2) comparing the derived *Vs* with reference soil NDVI from the ICRAF soil spectral library; and 3) validating the accuracy of FVC calculated from the pure NDVI values against field-measured FVC. Our findings reveal the urgent need for reliable *Vv* and *Vs* per pixel for large-area FVC production. Traditional statistical methods are impractical to achieve this goal due to their reliance on pure pixels. The MultiVI algorithm has proven to be a viable solution, yielding *Vv* and *Vs* with a more reliable spatial pattern and magnitude than statistical methods. The MultiVI *Vs* is closely aligned with soil NDVI across various soil types in China's mainland. Furthermore, the FVC estimated using the MultiVI *Vv* and *Vs* demonstrated improved accuracy in comparison to those derived from the statistical *Vv* and *Vs*, with RMSD values around 0.1 and $R^2$ values near 0.8 for all validation sites. Moreover, the products generated in this study show broad applicability across a variety of climate zones and soil types.

The MultiVI *Vv* and *Vs* maps provide essential parameters for FVC estimation using the widely adopted VI-based mixture model, which is known for its ease of use and reasonable accuracy. Hence, these derived *Vv* and *Vs* maps are anticipated to facilitate the estimation of fine-resolution, high-frequency FVC with reliable quality at large scales.

## 7 Data availability

The 30 m MultiVI *Vv* and *Vs* maps are available at https://zenodo.org/records/14060222 (Zhao et al., 2024). The *Vv* and *Vs* data are stored in GeoTIFF files using the WGS-84 (World Geodetic System 1984) coordinate system with UTM (Universal Transverse Mercator) projection. The data are categorized into tiles based on a latitude size of 5° and a longitude size of 6°. The file names consist of 18 characters following these rules: North-South latitude abbreviation (1 digit) + 6-degree zone number (2 digits) + "_" + starting latitude (2 digits) + "_" + product time (4 digits) + "_ " + resolution (3 digits) + "_ " + data attribute (2 digits). Additionally, each image contains a pure NDVI value band range from 0 to 100, with invalid values or water surfaces labelled as NaN. The 500 m MultiVI *Vv* and *Vs* maps are also publicly available at https://zenodo.org/records/15597968 (Zhao et al., 2024).

## 8 Author contribution

Xihan Mu, Wanjuan Song, and Tian Zhao designed the methodology. Wanjuan Song and Tian Zhao programmed the software and generated the data. Tian Zhao wrote the original draft, and Xihan Mu reviewed the manuscript. Hangqi Ren processed the data for the revised manuscript. Yuanyuan Wang, Yun Xie, Donghui Xie, and Guangjian Yan supervised the project.

## 9 Competing interests

The authors declare that they have no conflict of interest.

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
