# Peer review of "Normalized Difference Vegetation Index Maps of Pure Pixels over China's mainland for Estimation of Fractional Vegetation Cover"

_Earth System Science Data, 2024_

## Author Comment (AC1)

In the following, we addressed the specific points of the reviewers. Reviewer comments are black font and our responses are blue. We also use red highlights to mark changes in the revised manuscript.

We use codes to the Reviewers' comments, for example R1C2 means Reviewer 1 Comment 2.

**Reviewer #1:**

Review of Zhao et al.

This manuscript proposes a new approach to estimating Fractional Vegetation Cover (FVC) across China using the MultiVI algorithm, which integrates multiple remote sensing data. The results generally have good accuracy and spatial coherence, validated through field measurements. The manuscript is well written, and the methodology is well presented. The only major issue is that this dataset is for the year 2014.

Re: Thank you for your encouragement and affirmation. Your comments helped improve our manuscript. We revised our manuscript and responded to the comments and suggestions point by point as follows.

**General comments:**

**R1C1:** Limitation of single-year data. How representative can the use of single-year data (in 2014) be for the interannual variability in vegetation and soil properties? Why didn't the authors expand the methods to more recent years?

Re: **Done.** Thank you for your valuable comment. We newly generated Vv and Vs datasets for the years 2018 and 2022, and found that the values of Vv and Vs show very small differences across different years. To address your concern regarding the representativeness of using endmember values from a single year, we have conducted a supplementary analysis and revised the main text accordingly (Section 5), including the following points:

- 1) The NDVI values of pure vegetation (Vv) and bare soil pixels (Vs) are primarily influenced by factors such as plant species, soil types, climate, and moisture conditions, which typically remain stable unless disrupted by sudden events like fire or land cover conversion. This inherent stability in the land surface background implies that the estimation of FVC imposes relatively low demands on the temporal frequency of endmember calibration. Furthermore, we calculated the differences between the Vv and Vs obtained for 2022 and those from 2014. The results demonstrate that the endmember values show very small interannual differences. The supplementary analysis has been added in Section 5.
- 2) The field validation data used in this study span from 2012 to 2022, and the results show consistent and accurate FVC estimation across multiple years, indicating the broader applicability of the 2014-derived endmembers. Moreover, using one set of *Vv* and *Vs* to calculate FVC across multiple years demonstrated a reasonable

accuracy in many other studies (Oleson et al., 2000; Zhao et al., 2023; Donohue et al., 2025).

- 3) MODIS sensors are known to exhibit time-dependent signal degradation, which may lead to increased uncertainty in BRDF-derived products over time. To minimize the impact of sensor degradation, we selected data from the year 2014, when MODIS data quality was relatively stable and well-calibrated.
- 4) We have also generated additional *Vv* and *Vs* datasets for the years 2018 and 2022 using the same methodology. These datasets are currently being prepared and are expected to be published as a companion product with the next revised manuscript.

Newly added Reference: Donohue, R. J. and Renzullo, L. J.: An assessment of the accuracy of satellite-derived woody and grass foliage cover estimates for Australia, Aust. J. Bot., 73, BT24060, https://doi.org/10.1071/BT24060, 2025.

**Specific comments:**

**R1C2:** L27: should briefly introduce the reasons for using these three regions (e.g. for validation purposes), otherwise the readers will be confused as to why only compare to these regions.

Re: **Done.** Thank you for your suggestion. We have revised the abstract to clarify the rationale for selecting the three validation regions (P1: Line 28~29).

P1: Line 28~29

"These regions include typical arid and humid zones in China, facilitating the evaluation of the algorithm's performance under diverse climatic conditions."

**R1C3:** L30: 'free access' to 'publicly available' Re: **Done.** Thanks for your correction. We have replaced the phrase in the manuscript.

**R1C4: L30: should add what year is the data for**

Re: **Done.** Thanks for your reminder. We have clarified the year for the data. *P1: Line 30~31*

"The 30 m pure NDVI maps of 2014 are publicly available at https://zenodo.org/records/14060222 (Zhao et al., 2024)."

**R1C5: L93: remove 'flexibly'**

Re: Done. The word "flexibly" has been removed accordingly.

*P5: Line 93~94*

"These datasets can be applied to accurately calculate FVC at various resolutions on regional or national scales."

**R1C6:** L113: need more details about the choice of 55 and 60 degrees.

Re: **Done.** Thanks for your suggestion. A more detailed description of this angular configuration was provided in Section 3.1.1: "This selection is attributed to its minimal influence on  $G(\theta)$  and the high quality of angular remote sensing observations (Mu et al., 2018)." To enhance clarity, we have also cross-referenced this explanation at the point you mentioned (P5: Lines 113–115), allowing readers to easily locate the relevant details.

**P5: Line 113~115**

"All MCD43A1 data obtained in 2014 over China's mainland were used to reconstruct the ground surface reflectance of red and near-infrared (NIR) bands at viewing zenith angles (VZAs) of 55° and 60° (see Section 3.1.1 for more details on the choice of angular configuration)."

**R1C7:** L272: A moving window of 330x330m might oversimplify the spatial heterogeneity, how does it affect accuracy?

Re: **Done.** Thank you for your insightful comment. To avoid oversimplifying spatial heterogeneity, we have removed the  $330 \times 330$  m moving window when calculate the statistical *Vv* and *Vs*. Moreover, we recalculated the statistical *Vv* and *Vs* values pixel-by-pixel using a longer Landsat time series (2010–2020) to improve their temporal stability and representativeness, as suggested by Reviewer 2. Corresponding revisions have been made in Section 3.3.1.

**R1C8:** Figure 6: I suggest changing the colors by using darker colors to indicate larger differences (e.g. dark blue for -0.3~-0.2, light blue for -0.1~0)

Re: **Done.** Thanks for your advice and we have revised the Figure 6 accordingly to enhance its clarity.

**R1C9:** L335: why compare the mean (of MultiVI) with the median (NDVI)? Why not mean with mean or median with median?

Re: **Done.** Using different statistical measures can be misleading and we appreciate your reminder. Since boxplots typically use the median as the central tendency indicator, we have updated the comparison to consistently use the median for both MultiVI and Statistical *Vs*. The median and mean values of MultiVI and Statistical *Vs* across soil types are very close (with difference values less than 0.02 in most soil types), so the original conclusions remain unchanged. Corresponding updates have been made to Figure 7 and the related text descriptions.

"Figure 7: The boxplot of the soil NDVI from the ICRAF soil library for each soil type. Each boxplot features a central red line representing the median. The N above the box indicates the number of sampling plots for each soil type. The lower and upper edges of the box denote the 25th and 75th percentiles, respectively. The whiskers are extended to the most extreme data points excluding outliers. The blue and red lines denote the median values of the MultiVI *Vs* and statistical *Vs*, respectively."

**R1C10:** L348: add what 'the bias' represents (it is already in Figure 8 legend, better to have it in the main text).

Re: **Done.** Thanks for your reminder. We have added a sentence in the main text to clarify the meaning of 'the bias'.

**R1C11:** Figure 9: there seem to be seasonal patterns for some sites by eye, and it is worth further exploration.

Re: **Done.** Thank you for your comment. We have added descriptions of the seasonal patterns observed in Figure 9 to Section 4.3, and further discussed the underlying causes of these seasonal differences in the Section 5.

"Figure 9 shows that FVC errors are generally larger during early spring and winter, particularly at low FVC values. This seasonal pattern can be explained by the sensitivity of FVC estimation to NDVI values in the 0.2–0.4 range (Montandon and Small, 2008). In this interval, small errors in *Vs* can lead to systematic overestimation of FVC, especially over grasslands and shrublands. During peak growing seasons, when NDVI values exceed 0.7, the model estimates become more stable and less sensitive to endmember NDVI values."

**R1C12:** L457: usually invalid values should be marked as nan, not 0 to avoid confusion with actual 0 values.

Re: **Done.** We have updated the published data by marking invalid values as NaN instead of 0 to avoid confusion with actual zero values. The corresponding description in the manuscript has also been revised.

---

## Author Comment (AC2)

Response letter for 'ESSD-2024-535 Normalized Difference Vegetation Index Maps of Pure Pixels over China's mainland for Estimation of Fractional Vegetation Cover'.

In the following, we addressed the specific points of the reviewers. Reviewer comments are black font and our responses are blue. We also use red highlights to mark changes in the revised manuscript.

We use codes to the Reviewers' comments, for example R1C2 means Reviewer 1 Comment 2.
* * *
**Reviewer #2:**

Review of ESSD-2024535 Normalized Difference Vegetation Index Maps of Pure Pixels over China's mainland for Estimation of Fractional Vegetation Cover, by Zhao and others.

The authors have addressed an important issue in the use of NDVI for monitoring foliage cover. The end members of the linear transform from NDVI to cover need to be specified and this is commonly ignored. A robust method for routinely identifying these end members across diverse ecosystem types is needed, and is provided in this work. The $Vv$ and $Vs$ data generated will be valuable. The methods for generating the 500 m version of these variables are sound; the methods for downscaling these to 30 m need improvement. Further, the methods used by the authors for validating these surfaces are not robust and also need to be revised. I recommend a major revision.

Re: **Done.** Thank you for recognizing the effectiveness of our algorithm and for providing constructive suggestions regarding the downscaling logic. In response to the concerns about potential downscaling errors in the 30 m product and limitations in the validation methodology, we have made the following improvements:

1) To address the lack of clarity in the downscaling method, we have provided the complete set of downscale equations (Equation 9), and updated the corresponding textual description to enhance transparency and eliminate possible misunderstandings;

2) We have reorganized the description of the downscaling process, explicitly analyzed the potential errors introduced by the underlying assumption, and acknowledged the possible limitations of the approach. It is worth noting that validation results show good FVC estimation accuracy at 30 m resolution, indicating that the potential errors caused by the downscaling process are acceptable in practice;

3) We have improved the statistical method by using a longer time series (2010-2020) Landsat data to derive the endmember NDVI values, thereby enhancing its representativeness;

4) We have cited the recent work by Donohue and Renzullo (2025), which demonstrates that more sophisticated statistical approaches can also achieve

accurate FVC estimates. In addition, we revised our manuscript to avoid overgeneralized descriptions of the limitations of statistical methods.

**R2C1:** A significant concern I have is with the downscaling of *Vv* and *Vs*. The method for calculating 500 m *Vs* and *Vv* are sound and the 500 m data are an excellent product. The logic of the downscaling step, and uncertainty about how this downscaling was performed, significantly weakens the quality of the 30 m product. The downscaling introduces the assumption that *Vv* and *Vs* are the same within a given land cover type (line 249). This assumption rarely holds true as soil types (the main driver of *Vs* if one ignores the effects of soil moisture) can vary within single landcover types, or, conversely, different landcovers can share the same soil type. This assumption opens the authors up to the same criticism that they have applied to traditional statistical methods (line 395).

Re: **Done.** Thanks for your insightful comments. Aiming at analyzing and clarifying the potential uncertainty caused by downscaling step, we have reorganized the method (Section 3.2) and discussion (Section 5), including the following points:

1) We appreciate your affirmation of the 500 m *Vv* and *Vs* data. We also published the 500 m *Vs* and *Vv* as supplement, which can facilitate the coarse-resolution FVC estimation. The datalink has been added in Section 7.

2) Unlike traditional statistical methods that often assume a uniform *Vs* value for the same land cover type across large spatial extents (e.g., national or eco-regional scales), our downscaling approach applies this assumption only within a localized 3×3 window of 500 m MODIS pixels. This assumption may introduce uncertainty only when substantial soil type variation exists within the 3×3 MODIS window (1.5km × 1.5km). This implies that the error introduced by assuming homogeneity within the same land cover type is likely to be limited. The relevant description has been clarified in Section 3.2.

3) We fully acknowledge the dependence of *Vs* on soil type. However, the *Vs* values are influenced not only by mineral soil reflectance, but also by non-photosynthetic vegetation (NPV) and biological components such as mosses or lichens. A previous study reported that the NDVI difference between bare soil and NPV endmembers can reach up to 0.2 (Tian et al. 2021), indicating that *Vs* may vary even within the same soil type. Figure 7 further shows that the retrieved *Vs* values deviate from soil NDVI in humid regions, likely due to the influence of surface litter and biological residues. Land cover classification can partly account for such heterogeneity, we used land cover data as a practical proxy for disaggregating *Vs*. Corresponding revisions have been made in the manuscript to clarify this rationale and discuss potential uncertainties in Section 5.

4) Despite the simplifications involved, the comparison with 500 m results shows that the downscaled 30 m *Vs* values achieve comparable accuracy (Song et al., 2022). The statistical comparison shows that the downscaling process introduces minimal changes to the endmember values (Song et al., 2022). This suggests that the downscale process preserves the overall spectral characteristics of the original

MODIS-derived endmembers. Considering the increasing demand for high spatial and temporal resolution applications, we believe that providing 30 m endmember products is of practical significance. Corresponding explanations and references have been added to the manuscript in Section 5.

Newly added Reference: Tian, J., Su, S., Tian, Q., Zhan, W., Xi, Y., & Wang, N. (2021). A novel spectral index for estimating fractional cover of non-photosynthetic vegetation using near-infrared bands of Sentinel satellite. *International Journal of Applied Earth Observations and Geoinformation*, 101, 102361. https://doi.org/10.1016/j.jag.2021.102361

**R2C2:** Further, it is difficult to understand how this downscaling was performed as the methods do not currently describe a proper unmixing method. Equation 8 apportions *Vs* (or *Vv*) solely according to landcover type proportion, regardless of which landcover type occupies that proportion. As currently described, for a hypothetical 500 m pixel with a calculated *Vs* value and which has 10% area of forest and 10% bare ground (in the surrounding 3x3 window), the method would apportion the same *Vs* value to the forest and bare pixels. Can the authors better explain the method used?

Re: **Done.** Thank you for raising this important point. We have clarified the disaggregation process in the revised manuscript and provide the following step-by-step explanation to address your concern:

1) For each target MODIS pixel, we define a 3×3 window centered on it (i.e., covering 9 MODIS pixels). We assume that within this local window, each land cover type $k$ has a consistent endmember value $V_{v,k}$ or $V_{s,k}$.

2) We construct a system of linear equations (as illustrated in Equation 9), where the known variables are the MODIS-scale $V_v$ (or $V_s$) values for the 9 pixels and the land cover fractions $f_{k,x,y}$ of each type within each MODIS pixel. The unknowns are the land cover-specific values $V_{v,k}$ (or $V_{s,k}$) within the window.

$$
\begin{cases}
V_{v,\text{modis},x-1,y-1} = \sum_{k=1}^{m} f_{k,x-1,y-1} V_{v,k,x,y} \\
V_{v,\text{modis},x-1,y} = \sum_{k=1}^{m} f_{k,x-1,y} V_{v,k,x,y} \\
V_{v,\text{modis},x-1,y+1} = \sum_{k=1}^{m} f_{k,x-1,y+1} V_{v,k,x,y} \\
V_{v,\text{modis},x,y-1} = \sum_{k=1}^{m} f_{k,x,y-1} V_{v,k,x,y} \\
V_{v,\text{modis},x,y} = \sum_{k=1}^{m} f_{k,x,y} V_{v,k,x,y} \\
V_{v,\text{modis},x,y+1} = \sum_{k=1}^{m} f_{k,x,y+1} V_{v,k,x,y} \\
V_{v,\text{modis},x+1,y-1} = \sum_{k=1}^{m} f_{k,x+1,y-1} V_{v,k,x,y} \\
V_{v,\text{modis},x+1,y} = \sum_{k=1}^{m} f_{k,x+1,y} V_{v,k,x,y} \\
V_{v,\text{modis},x+1,y+1} = \sum_{k=1}^{m} f_{k,x+1,y+1} V_{v,k,x,y}
\end{cases}
\tag{9}
$$

3) We solve this overdetermined system to obtain the optimal values $V_{v,k}$ or $V_{s,k}$. The value corresponding to each land cover type in the center MODIS pixel is then assigned to all the 30 m pixels within that MODIS pixel that share the same land cover type.

4) This 3×3 window is then moved across the MODIS grid to repeat the estimation for each MODIS pixel.

It can be seen that only in the rare case where all the nine MODIS pixels in a 3×3 window have identical land cover proportions (e.g., the same ratio of forest to bare ground), the resulting estimates would be the same. To avoid potential misunderstanding, we have revised the text to clarify the downscaling logic and provided a complete formulation of the equations used in the method (see revised Section 3.2 and Equation 9).

**R2C3:** I have two significant concerns about the data used to validate/assess their products. The first is the rather unsophisticated way the authors have applied the 'statistical' method. They have only used 3 years of data to derive statistics about $Vv$ and $Vs$. What if that period was continually wet, or continually dry, or was fire affected? The derived statistics cannot be assumed to be representative of that site. The authors have the ability to use a much longer time series and should do so. Also, the authors have applied the method with the expectation that it will work everywhere, which it is known not to. The method cannot return reliable $Vs$ values in heavily vegetated areas nor $Vv$ in sparsely vegetated areas. While the authors acknowledge this in the conclusion, this knowledge hasn't been applied in their design of the derivation of the statistically derived $Vs$ $Vv$ data. And so it is no surprise that this product performs poorly in these respective situations. This led the authors to conclude that (line 442) "Traditional statistical methods are impractical to achieve this goal due to their reliance on pure pixels."
This is not universally true. More sophisticated implementations of the statistical method can be quite effective. Can the authors at least provide some more context to the reader about the simplicity of their approach relative to alternative approaches? Or maybe the authors could restrict the application of their statistical method to where it is known to be valid and hence avoid reporting values where it quite rightly doesn't work. None of this will change the excellent result that the multi-VI method is superior.
Re: **Done.** Thank you for your thorough and constructive comments. We have carefully revised the manuscript to address your concerns regarding the statistical method used for deriving $Vv$ and $Vs$.

1) To improve the robustness of the statistical endmembers, we have recalculated $Vv$ and $Vs$ pixel by pixel using a longer time series of Landsat data. Specifically, the maximum and minimum NDVI values over the period 2010–2020 were used to represent $Vv$ and $Vs$, respectively. After this update, the rationality of the statistical endmembers has improved: $Vv$ values increased in humid region, while $Vs$ values decreased in arid area. The corresponding description has been updated in Section 2.2.1.

2) We clarified that the statistical method used in this study yields reasonable FVC estimation in most regions, except in evergreen forest areas and extremely arid zones. The updated content demonstrates that, outside of these extreme regions, the statistical endmembers provide reliable FVC (Figure 9). The practicability of the statistical method has now been explicitly stated, and the corresponding analysis has been supplemented in Sections 4.2 and 4.3.

3) We fully acknowledge the practicality of statistical methods, especially their simplicity and reasonable accuracy in suitable area. In fact, the MultiVI model proposed in this study incorporates statistical endmembers as boundary for inversion. We have revised the manuscript in the introduction and discussion to avoid overgeneralized or dismissive statements regarding statistical methods, and have expanded our discussion to better reflect their strengths and appropriate use cases. In addition, we have cited the recent work by Donohue and Renzullo (2025), which demonstrates that improved statistical implementations—when used with appropriate constraints—can achieve reliable results.

**R2C4:** The second concern I have about the data used to validate/assess their products relates to how the field data at Heihe were derived. In scaling the field observations from 10 x 10 m to 90 x 90 m, the authors have effectively turned the field observations into a modelled product with its own errors. I would expect that a direct comparison between the 10 m field data and the 30 m *Vs Vv* data would provide a more robust comparison than upscaling the field data.

Re: **Done.** Thank you for your insightful comment. Following your suggestion, we have revised the validation approach for the Heihe site. Specifically, we directly compared the $10 \times 10$ m field-measured FVC with the 30 m MultiVI and statistical FVC estimates, instead of upscaling the field data. The updated validation results are now presented in Figure 9.

**R2C5:** One last comment is that some recent work is of direct relevance to this Mutil-VI paper (Donohue and Renzullo, 2025; https://doi.org/10.1071/BT24060). I expect this would have been published after the current manuscript's submission; however, it may be of interest to the authors. In making this statement I should also disclose that this is my paper (it's Randall Donohue here).

Re: **Done.** Thank you for sharing your recent work and for disclosing your authorship. We appreciate the valuable contribution of your study, which proposes improvements to traditional statistical methods and demonstrates their effectiveness in estimating FVC over Australia. We have cited this reference in the revised manuscript to acknowledge that the statistical method adopted in our study is relatively simple and that more advanced implementations, such as yours, can achieve high estimation accuracy.

Newly added Reference: Donohue, R. J. and Renzullo, L. J.: An assessment of the accuracy of satellite-derived woody and grass foliage cover estimates for Australia, Aust. J. Bot., 73, BT24060, https://doi.org/10.1071/BT24060, 2025.

**R2C6:** Lines 42 and 49. The VI-based mixture model referred by the authors is specifically the NDVI-based mixture model. It is not a generic model that can use *any* vegetation index.

Re: **Done.** Thank you for your helpful comment. While the current study focuses on the NDVI-based mixture model, we would like to clarify that both the traditional VI-based

mixture model and the proposed MultiVI method are applicable to other vegetation indices, such as EVI. Previous studies applying MultiVI to EVI have also reported high estimation accuracy (Song et al., 2022a). To avoid confusion, we have added a sentence in the introduction to clarify that the model framework is not limited to NDVI.

**R2C7:** Line 173. Doesn't look like the UAV data were used for anything at the Hebei site. Do they need to be mentioned at all?
Re: **Done.** Thank you for pointing this out. At the Hebei site, the grassland data were indeed acquired using UAV, and these data were included in the validation analysis. We have revised the text to clarify the role of the UAV data and avoid potential confusion.

**R2C8:** Line 200. It is a misconception that the NDVI has a saturation effect. When compared to foliage cover (which it what is has been shown to be linearly related to) there is no 'saturation'. This misconception arises when NDVI is incorrectly expected to bear some relationship with leaf area.
Re: **Done.** Thanks for your suggestion. Instead of using the term "saturation," we have revised the text to emphasize the nonlinear relationship between NDVI and FVC. Although NDVI and FVC exhibit an approximately linear relationship in pure pixel assumptions, several studies have shown that, in practice, NDVI often displays a nonlinear response to FVC in mixed pixels due to the influence of multiple factors (Montandon and Small, 2008). These include soil background variability, sub-pixel shadow fractions, viewing geometry, terrain effects, and especially the spatial scale of observation (Mu et al. 2024). We revised the sentence in the manuscript to avoid misunderstanding.
Newly added Reference: Mu, X., Yang, Y., Xu, H., Guo, Y., Lai, Y., McVicar, T. R., Xie, D., & Yan, G. (2024). Improvement of NDVI mixture model for fractional vegetation cover estimation with consideration of shaded vegetation and soil components. Remote Sensing of Environment, 314, 114409. https://doi.org/10.1016/j.rse.2024.114409

**R2C9:** Line 229. Calling the values derived from a single year (2014) the 'historical' values is counterintuitive. They are not representative of site history.
Re: **Done.** Re: Thank you for your comment. To avoid the misunderstanding, we have revised the wording to clarify that the minimum and maximum NDVI values were derived from all available observations within the year 2014.

**R2C10:** Line 229. How much does using statistics derived from only one year of data (2014) limit the accuracy of the method when applied to other years? I would think it important to derive these 'historical' values from as long a time series as possible (which would be 23 or so years for MODIS).

Re: **Done.** Thank you for your comment. In response, we have revised the manuscript and addressed this issue from the following perspectives:

1) We acknowledge that assessing the representativeness of single-year data is important. As detailed in our response to R1C1, we conducted a supplementary analysis comparing $Vv$ and $Vs$ from 2014 with those from 2018 and 2022. The results show minimal interannual differences. We also clarified that the NDVI values of pure vegetation and bare soil pixels are generally stable across years unless affected by abrupt disturbances. Please refer to our response to R1C1 for full justification and supporting evidence.

2) The MultiVI algorithm estimates $Vv$ and $Vs$ by solving equations derived from two angular observations with significantly different NDVI values. As long as the differences of angular observations are sufficient, a valid solution can be obtained. A single year of MODIS data recorded a complete vegetation growth cycle, ensuring the availability of angular NDVI pairs with sufficient contrast for reliable inversion. Our experiments show that introducing too many angular observations can lead to overfitting, which degrades the estimation accuracy of $Vv$ and $Vs$. Therefore, selecting a representative set of well-separated angular observations from one year is an effective strategy to ensure solution quality while avoiding overfitting.

---

## Author Response (AR1)

Response letter for 'essd-2024-535 Normalized Difference Vegetation Index Maps of Pure Pixels over China's mainland for Estimation of Fractional Vegetation Cover'.

In the following, we address the specific points raised by the reviewers. Reviewer comments are presented in black font, while our responses are in blue. Additionally, we have used red highlights to mark changes made in the revised manuscript. We use codes for the reviewers' comments, for example, R1C2 refers to Reviewer 1, Comment 2.

We have thoroughly revised the entire manuscript, addressing the following key points:

1) New $Vv$ and $Vs$ datasets for 2018 and 2022 were generated to assess interannual variability, thereby confirming the reliability of the 2014 data (see R1C1, R2C10).

2) We recalculated the statistical endmembers using Landsat data from 2010 to 2020, which resulted in enhanced robustness and accuracy of the validation data (see R2C3, R1C7).

3) Detailed downscaling equations and unmixing logic have been incorporated to enhance methodological transparency and identify potential sources of error (see R2C1, R2C2).

4) We minimized unnecessary upscaling in field data validation and incorporated an analysis of seasonal FVC errors to enhance interpretability and provide practical insights (see R1C11, R2C4).
* * *
**Reviewer #1:**

This manuscript proposes a new approach to estimating Fractional Vegetation Cover (FVC) across China using the MultiVI algorithm, which integrates multiple remote sensing data. The results generally have good accuracy and spatial coherence, validated through field measurements. The manuscript is well written, and the methodology is well presented. The only major issue is that this dataset is for the year 2014.

Re: Thank you for your encouragement and affirmation. Your comments helped improve our manuscript. We have analyzed the limitations of the single-year data and generated new datasets for recent years, which will be released in subsequent revisions. Details of this analysis are provided in our response to R1C1. We have revised our manuscript and responded to the comments and suggestions point by point as follows.

**General comments:**

**R1C1:** Limitation of single-year data. How representative can the use of single-year data (in 2014) be for the interannual variability in vegetation and soil properties? Why didn't the authors expand the methods to more recent years?
Re: **Done.** Thank you for your valuable comment. We have newly generated the *Vv* and *Vs* datasets for the years 2018 and 2022, and found that the values of *Vv* and *Vs* show very small differences across different years. To address your concern regarding the representativeness of using endmember values from a single year, we have conducted a supplementary analysis, including the following points:

1) The NDVI values of pure vegetation (*Vv*) and bare soil pixels (*Vs*) are primarily influenced by factors such as plant species, soil types, climate, and moisture conditions, which typically remain stable unless disrupted by sudden events like fire or land cover conversion. This inherent stability in the land surface background implies that the estimation of FVC imposes relatively low demands on the temporal frequency of endmember calibration. Furthermore, we calculated the differences between the *Vv* and *Vs* obtained for 2018 and those from 2014. The results demonstrate that the endmember values show small interannual differences (Figure S1). This analysis has been added as supplementary material.

2) The field validation datasets used in this study span from 2012 to 2022. The results show consistent and accurate estimation of FVC across multiple years, indicating the broad applicability of the endmembers derived in 2014. Moreover, using one set of *Vv* and *Vs* to calculate FVC across multiple years demonstrated a reasonable accuracy in many other studies (Zhao et al., 2023; Donohue et al., 2025).

3) We have also generated additional *Vv* and *Vs* datasets for the years 2018 and 2022 using the same methodology. These datasets are currently being prepared and are expected to be published as a companion product alongside the next revised manuscript.

Newly added Reference:
Donohue, R. J. and Renzullo, L. J.: An assessment of the accuracy of satellite-derived woody and grass foliage cover estimates for Australia, Aust. J. Bot., 73, BT24060, https://doi.org/10.1071/BT24060, 2025.

*Supplementary materials:*

"The differences in NDVI values for pure vegetation ($Vv$) and bare soil ($Vs$) between 2014 and 2018, as derived from the MultiVI algorithm, are illustrated in Figure S1. In most regions, the interannual variations in endmember NDVI values from 2014 to 2018 fall within ±0.1.

The $Vv$ and $Vs$ are primarily influenced by factors such as plant species, soil types, climate, and moisture conditions, which typically remain stable unless disrupted by sudden events like fire or land cover conversion. This inherent stability in the land surface background implies that the estimation of FVC imposes relatively low demands on the temporal frequency of endmember calibration.

Furthermore, the field validation datasets used in this study span from 2012 to 2022, achieving consistent accuracy in FVC estimation across multiple years with the 2014-derived endmembers. This finding underscores the broad applicability of the 2014-derived endmembers. Additionally, employing a single set of $Vv$ and $Vs$ to calculate FVC over multiple years has demonstrated reasonable accuracy in several other studies (Zhao et al., 2023; Donohue and Renzullo, 2025)."

[Figure]

[Figure]

**Figure S1: The difference maps of MultiVI NDVI between 2014 and 2018 for pure vegetation and bare soil.**

**Specific comments:**

**R1C2:** L27: should briefly introduce the reasons for using these three regions (e.g. for validation purposes), otherwise the readers will be confused as to why only compare to these regions.
Re: **Done.** Thank you for your suggestion. We have revised the abstract to clarify the rationale for selecting the three validation regions.
*P1: Line 32~33*
"These regions include typical arid and humid zones, facilitating the evaluation of the algorithm's performance under diverse climatic conditions."

**R1C3:** L30: 'free access' to 'publicly available'

Re: **Done.** Thanks for your correction. We have replaced the phrase in the manuscript.

*P2: Line 35~36*

"The 30 m pure NDVI maps of 2014 are publicly available at https://zenodo.org/records/14060222 (Zhao et al., 2024)."

**R1C4:** L30: should add what year is the data for

Re: **Done.** Thanks for your reminder. We have clarified the year for the data.

*P2: Line 35~36*

"The 30 m pure NDVI maps of 2014 are publicly available at https://zenodo.org/records/14060222 (Zhao et al., 2024)."

**R1C5:** L93: remove 'flexibly'

Re: **Done.** The word "flexibly" has been removed accordingly.

*P5: Line 100~101*

"These datasets can be applied to accurately calculate FVC at various resolutions on regional or national scales."

**R1C6:** L113: need more details about the choice of 55 and 60 degrees.

Re: **Done.** Thanks for your suggestion. A more detailed description of this angular configuration was provided in Section 3.1.1: "This selection is attributed to its minimal influence on $G(\theta)$ and the high quality of angular remote sensing observations (Mu et al., 2018)." To enhance clarity, we have also cross-referenced this explanation at the point you mentioned, allowing readers to easily locate the relevant details.

*P5: Line 120~122*

"All MCD43A1 data obtained in 2014 over China's mainland were used to reconstruct the ground surface reflectance of red and near-infrared (NIR) bands at viewing zenith angles (VZAs) of 55° and 60° (see Section 3.1.1 for further details on the selection of angular configuration)."

Details on the selection of angular configuration in Section 3.1.1:

*P10: Line 227~229*

"The combination of 55° and 60° in the forward viewing directions was identified as the optimal angular configuration. This selection is attributed to its minimal influence on $G(\theta)$ and the high quality of angular remote sensing observations (Mu et al., 2018)."

**R1C7:** L272: A moving window of 330x330m might oversimplify the spatial heterogeneity, how does it affect accuracy?

Re: **Done.** Thank you for your insightful comment. To avoid oversimplifying spatial heterogeneity, we have removed the $330 \times 330$ m moving window when calculating the statistical *Vv* and *Vs*. Moreover, we recalculated the statistical *Vv* and *Vs* values pixel-by-pixel using a longer Landsat time series (2010–2020) to improve their temporal

stability and representativeness, as suggested by Reviewer 2. Corresponding revisions have been made in Section 3.3.1.

*P13: Line 280~282*

The statistical method utilized Landsat 8 data from 2010 to 2020, processed on the Google Earth Engine (GEE) platform. The pixel-wise maximum and minimum NDVI values from 2010 to 2020 were set as the *Vv* and *Vs*, respectively.

**R1C8:** Figure 6: I suggest changing the colors by using darker colors to indicate larger differences (e.g. dark blue for -0.3~-0.2, light blue for -0.1~0)

Re: **Done.** Thanks for your advice and we have revised the Figure 6 accordingly to enhance its clarity.

[Figure]

[Figure]

Figure 6: The difference maps between the MultiVI NDVI and statistical NDVI for pure vegetation and bare soil.

**R1C9:** L335: why compare the mean (of MultiVI) with the median (NDVI)? Why not mean with mean or median with median?

Re: **Done.** Utilizing different statistical measures can lead to misleading interpretations, and we appreciate your reminder. Since boxplots typically employ the median as the indicator of central tendency, we have revised the comparison to consistently use the median for both MultiVI and statistical *Vs*. As suggested by Reviewer 2, we utilized a longer time series of Landsat data (2010-2020) to calculate statistical *Vs* and compare them with the soil NDVI values from the soil spectral library. Based on the updated results presented in Figure 7, we have reorganized the related text descriptions in Section 4.2.

*P16: Line 346~355*

"Figure 7 shows the MultiVI *Vs* and statistical *Vs* in comparison to the soil NDVI derived from the ICRAF soil library. For Alfisols and Semi-Luvisols, the median of the MultiVI *Vs* closely aligns with the median NDVI of the corresponding soil samples. In contrast, the median statistical *Vs* tend to slightly overestimate the soil NDVI. For Desert soils and Skeletol primitive soils, the median of the statistical *Vs* is closer to the soil NDVI, while the MultiVI *Vs* exhibit a slight underestimation. Both the median values of the MultiVI and the statistical *Vs* are lower than the median NDVI for Dark Semi-hydromorphic soils, with a bias of approximately 0.1. Soil samples of Anthrosols

and Ferralisols are primarily distributed in the humid, densely vegetated regions of southeastern China, which results in relatively high NDVI values. For these soil types, the MultiVI *Vs* show an overestimation when compared to the median NDVI values, with biases of approximately 0.15 for Anthrosols and 0.2 for Ferralisols. For Ferralisols, this overestimation is more pronounced in the statistically derived *Vs*, with biases exceeding 0.3."

[Figure]

**Figure 7: The boxplot of the soil NDVI from the ICRAF soil library for each soil type. Each boxplot features a central red line representing the median. The N above the box indicates the number of sampling plots for each soil type. The lower and upper edges of the box denote the 25th and 75th percentiles, respectively. The whiskers are extended to the most extreme data points, excluding outliers. The blue and red lines denote the median values of the MultiVI *Vs* and statistical *Vs*, respectively.**

**R1C10:** L348: add what 'the bias' represents (it is already in Figure 8 legend, better to have it in the main text).
Re: **Done.** Thanks for your reminder. We have added a sentence in the main text to clarify the meaning of 'the bias'.
*P17: Line 362~363*
"The FVC bias represents the difference between the MultiVI FVC (Figure 8a) or statistical FVC (Figure 8b) and the reference FVC."

**R1C11:** Figure 9: there seem to be seasonal patterns for some sites by eye, and it is worth further exploration.
Re: **Done.** Thank you for your comment. We have added descriptions of the seasonal patterns observed in Figure 9 to Section 4.3, and further discussed the underlying causes of these seasonal differences in Section 5.
*P19: Line 396~401*
"Figure 9 also reveals seasonal patterns in the errors of FVC derived from the VI-based

mixture model. At the Hebei site, larger deviations from the 1:1 line are observed in April and October, indicating greater uncertainty during the early growth and senescence stages. At the Heihe site, the FVC shows the largest error in June, particularly when 0.2<FVC<0.5. In the Three Gorges Reservoir area, most significant FVC errors occur during winter months (January to February) and summer months (July to September). The seasonal patterns in the errors of the statistical FVC are generally more pronounced than those of the MultiVI FVC."

[Figure]

**Figure 9: Scatterplots of the MultiVI FVC and the statistical FVC versus the field-measured FVC. N is the number of samples.**

*P21: Line 440~446*
"The errors in the FVC estimated from the VI-based mixture model vary across seasons and are more pronounced during periods of low vegetation cover. Figure 9 shows that FVC errors are generally larger in early spring and winter, particularly for low FVC values. This seasonal pattern can be explained by the sensitivity of FVC estimation to

NDVI values in the 0.2–0.4 range (Montandon and Small, 2008). Within this interval, small errors in $Vs$ can lead to systematic overestimations of FVC, especially over grasslands and shrublands. During peak growing seasons, when NDVI values exceed 0.7, the model estimates become more stable and less sensitive to endmember NDVI values. Furthermore, the saturation of FVC estimates, resulting from an underestimation of $Vv$ and an overestimation of $Vs$, typically influences winter and summer periods (Figure 9f)."

**R1C12:** L457: usually invalid values should be marked as nan, not 0 to avoid confusion with actual 0 values.
Re: **Done.** We have updated the published data by marking invalid values as NaN instead of 0 to avoid confusion with actual zero values. The corresponding description in the manuscript has also been revised.
*P22: Line 489~491*
"Additionally, each image contains a pure NDVI value band range from 0 to 100, with invalid values or water surfaces labelled as NaN."
* * *
**Reviewer #2:**

The authors have addressed an important issue in the use of NDVI for monitoring foliage cover. The end members of the linear transform from NDVI to cover need to be specified and this is commonly ignored. A robust method for routinely identifying these end members across diverse ecosystem types is needed, and is provided in this work. The $Vv$ and $Vs$ data generated will be valuable. The methods for generating the 500 m version of these variables are sound; the methods for downscaling these to 30 m need improvement. Further, the methods used by the authors for validating these surfaces are not robust and also need to be revised. I recommend a major revision.

Re: **Done.** Thank you for recognizing the effectiveness of our algorithm and for providing constructive suggestions regarding the downscaling logic and validation methods. The $Vs$ is influenced not only by soil types, but also by non-photosynthetic vegetation (NPV) and other biological components, contributing to its spatial variability. The adopted downscaling strategy based on the land cover products can partly explain this spatial variation. The adopted downscaling strategy, which incorporates land cover products, can partially account for this heterogeneity. In the revised manuscript, we have reorganized the downscaling procedures and provided a detailed analysis of the associated uncertainties. Notably, the resulting 30 m endmember products exhibit satisfactory accuracy.

In response to the concerns about potential downscaling errors in the 30 m product and the limitations of the validation methodology, we have made the following improvements:

1) To clarify the downscaling method, we have provided the complete set of downscaling equations (Equation 8) and updated the corresponding textual explanation to enhance clarity and prevent potential misunderstandings (see response to R2C1 and R2C2);

2) We have reorganized the description of the downscaling process, explicitly analyzed the potential errors introduced by the underlying assumptions, and acknowledged the possible limitations of the approach (see response to R2C1 and R2C2). Notably, the validation results demonstrate a good accuracy in FVC estimation at a 30 m resolution, indicating that the potential errors arising from the downscaling process are acceptable in practice;

3) We have improved the statistical method by using a longer time series of Landsat data (2010-2020) to derive the endmember NDVI values. The corresponding validation results (Section 4) have also been updated, indicating improved accuracy of the newly generated statistical endmembers (see response to R2C3);

4) We have cited the recent work by Donohue and Renzullo (2025), which demonstrates that more sophisticated statistical approaches can also yield accurate FVC estimates. In addition, we revised our manuscript to avoid overgeneralized descriptions of the limitations of statistical methods and to acknowledge their strengths (see response to R2C5).

**R2C1:** A significant concern I have is with the downscaling of *Vv* and *Vs*. The method for calculating 500 m *Vs* and *Vv* are sound and the 500 m data are an excellent product. The logic of the downscaling step, and uncertainty about how this downscaling was performed, significantly weakens the quality of the 30 m product. The downscaling introduces the assumption that *Vv* and *Vs* are the same within a given land cover type (line 249). This assumption rarely holds true as soil types (the main driver of *Vs* if one ignores the effects of soil moisture) can vary within single landcover types, or, conversely, different landcovers can share the same soil type. This assumption opens the authors up to the same criticism that they have applied to traditional statistical methods (line 395).

Re: **Done.** Thank you for your insightful comments. In order to analyze and clarify the potential uncertainties caused by the downscaling step, we have reorganized the methodology (Section 3.2) and the discussion (Section 5), including the following points:

1) We appreciate your confirmation of the 500 m *Vv* and *Vs* data. Additionally, we also published the 500 m *Vs* and *Vv* as a supplement, which can facilitate the coarse-resolution FVC estimation. The data link has been included in Section 7.

*P22: Line 490~491*

"The 500 m MultiVI *Vv* and *Vs* maps are also publicly available at https://zenodo.org/records/15597968 (Zhao et al., 2024)."

2) Unlike traditional statistical methods that often assume a uniform *Vs* value for the same land cover type across large spatial extents (e.g., national or eco-regional scales), our downscaling approach applies this assumption only within a localized 3×3 window of 500 m MODIS pixels. This assumption may introduce uncertainty only when significant soil type variation exists within the 3×3 MODIS window (1.5 km × 1.5 km). Consequently, the error introduced by assuming homogeneity within the same land cover type is likely to be minimal. The relevant description has been clarified in Section 3.2.

*P12: Line 256~259*

"It was assumed that the same land cover type within each MODIS pixel was assigned the same *Vv* and *Vs* values. The 500 m *Vv* and *Vs* were considered as linear combinations of the 30 m *Vv* and *Vs* values within that MODIS pixel, with weights determined by the areal proportions of land cover types."

3) We fully acknowledge the dependence of *Vs* on soil type. However, the *Vs* values are influenced not only by mineral soil reflectance, but also by non-photosynthetic vegetation (NPV) and biological components such as mosses or lichens. A previous study reported that the NDVI of NPV endmembers is generally higher than that of bare soil, with differences reaching up to 0.4 in some cases (Tian et al., 2021), indicating that *Vs* may vary even within the same soil type. Figure 7 further shows that the retrieved *Vs* values deviate from soil NDVI in humid regions, likely due to the influence of surface litter and biological residues. Since land cover classification can partly account for such heterogeneity, we used land cover data as

the basis for disaggregating *Vs*. Corresponding revisions have been made in the manuscript to clarify this rationale and discuss potential uncertainties in Section 5.

*P20: Line 417~421*

"In practice, *Vs* is influenced not only by mineral soil properties but also by non-photosynthetic vegetation (NPV), biological residues, and surface litter. A previous study reported that the NDVI of NPV endmembers is generally higher than that of bare soil, with differences reaching up to 0.4 in some cases (Tian et al., 2021). This suggests that *Vs* may vary even within the same soil type. Land cover classification can partially account for such heterogeneity, and land cover data were utilized as the basis for disaggregating 500 m *Vs* to a 30 m resolution in this study."

4) Despite the simplifications involved, the comparison with the 500 m results shows that the downscaled 30 m *Vs* values achieve comparable accuracy (Song et al., 2022b). The statistical comparison shows that the downscaling process introduces minimal changes to the endmember values (Song et al., 2022b). This suggests that the downscale process preserves the overall spectral characteristics of the original MODIS-derived endmembers. Considering the increasing demand for high spatial and temporal resolution applications, we believe that providing 30 m endmember products is of practical significance. Corresponding explanations and references have been added to the manuscript in Section 5.

*P21: Line 452~459*

"The downscaling procedure integrates a 30 m land cover product, which introduces finer spatial detail to describe the sub-pixel heterogeneity within MODIS pixels. Our downscaling approach applies the assumption that the 30 m pixels of the same land cover type share constant *Vv* and *Vs* values within a localized 3×3 window of 500 m MODIS pixels. This assumption may introduce uncertainty when significant soil type variation exists within the 3×3 MODIS window (1.5 km × 1.5 km). Despite the simplifications involved, the comparison with the 500 m results shows that the downscaled 30 m *Vs* values achieve comparable accuracy (Song et al., 2022b). The downscaling process introduces minimal changes to the endmember values (Song et al., 2022b). Considering the increasing demand for high spatial and temporal resolution applications, providing 30 m endmember products is of practical significance."

Newly added Reference:

Tian, J., Su, S., Tian, Q., Zhan, W., Xi, Y., and Wang, N.: A novel spectral index for estimating fractional cover of non-photosynthetic vegetation using near-infrared bands of Sentinel satellite, International Journal of Applied Earth Observation and Geoinformation, 101, 102361, https://doi.org/10.1016/j.jag.2021.102361, 2021.

**R2C2:** Further, it is difficult to understand how this downscaling was performed as the methods do not currently describe a proper unmixing method. Equation 8 apportions *Vs* (or *Vv*) solely according to landcover type proportion, regardless of which landcover type occupies that proportion. As currently described, for a hypothetical 500 m pixel with a calculated *Vs* value and which has 10% area of forest and 10% bare ground (in

the surrounding 3x3 window), the method would apportion the same *Vs* value to the forest and bare pixels. Can the authors better explain the method used?

Re: **Done.** Thank you for raising this important point. We have clarified the downscaling process in the revised manuscript and provided the following step-by-step explanation to address your concern:

1) For each target MODIS pixel, we define a 3×3 window centered on it (i.e., covering 9 MODIS pixels). We assume that within this local window (1.5 km × 1.5 km), each land cover type *i* has a spatially consistent endmember value $V_{v,i}$ or $V_{s,i}$.

2) We construct a system of linear equations (as illustrated in Equation 9), where the known variables are the MODIS *Vv* (or *Vs*) values for the 9 pixels and the land cover fractions $P_{i,x,y}$ for each land type in each MODIS pixel. The unknowns are the land-cover-specific endmember values $V_{v,i}$ (or $V_{s,i}$) within the window.

3) We solve this overdetermined system (up to 9 equations for ≤ 7 land cover types) using least-squares optimization to derive unique estimates of $V_{v,i}$ or $V_{s,i}$ within the specific window. These estimated values are then assigned to all 30 m pixels of each land cover type *i* within the central MODIS pixel at $(x,y)$.

4) This 3×3 window is moved across the MODIS grid to estimate the value for each MODIS pixel.

It can be seen that only in the rare case where all nine MODIS pixels within a 3×3 window have identical land cover proportions (e.g., the same ratio of forest to bare ground) will the system generate identical estimates for those land cover types. To avoid any potential misunderstandings, we have revised the main text to clarify the downscaling logic and have provided a complete formulation of the equations used in the method (see the revised Section 3.2 and Equation 9).

*P12: Line 256~276*

"The 500 m *Vv* and *Vs* were downscaled to a 30 m resolution using NDVI unmixing to facilitate fine-scale FVC estimation. It was assumed that the same land cover type within each MODIS pixel was assigned the same *Vv* and *Vs* values. The 500 m *Vv* and *Vs* were considered as linear combinations of the 30 m *Vv* and *Vs* values within that MODIS pixel, with weights determined by the areal proportions of land cover types.

The downscaling process utilized seven land cover types from the GlobeLand30 product, specifically four vegetation classes (cultivated land, forest, grassland, shrubland, and tundra), a grouped water surface category (wetland, water body, and permanent snow and ice), bare land, and artificial surfaces. A 3 × 3 sliding window with a step size of one MODIS pixel was employed to construct an overdetermined system of unmixing equations for the MODIS pixel at $(x, y)$, as shown in Eq. (8) for downscaling *Vv*:

$$
\begin{cases}
V_{v,\,\mathrm{modis},x-1,y-1} = \sum_{i=1}^{n} P_{i,x-1,y-1} V_{v,i,x,y} \\
V_{v,\,\mathrm{modis},x-1,y} = \sum_{i=1}^{n} P_{i,x-1,y} V_{v,i,x,y} \\
V_{v,\,\mathrm{modis},x-1,y+1} = \sum_{i=1}^{m} P_{i,x-1,y+1} V_{v,i,x,y} \\
V_{v,\,\mathrm{modis},x,y-1} = \sum_{i=1}^{n} P_{i,x,y-1} V_{v,i,x,y} \\
V_{v,\,\mathrm{modis},x,y} = \sum_{i=1}^{n} P_{i,x,y} V_{v,i,x,y} \\
V_{v,\,\mathrm{modis},x,y+1} = \sum_{i=1}^{n} P_{i,x,y+1} V_{v,i,x,y} \\
V_{v,\,\mathrm{modis},x+1,y-1} = \sum_{i=1}^{n} P_{i,x+1,y-1} V_{v,i,x,y} \\
V_{v,\,\mathrm{modis},x+1,y} = \sum_{i=1}^{n} P_{i,x+1,y} V_{v,i,x,y} \\
V_{v,\,\mathrm{modis},x+1,y+1} = \sum_{i=1}^{n} P_{i,x+1,y+1} V_{v,i,x,y}
\end{cases} , \tag{8}
$$

where $V_{v,\,\mathrm{modis},x,y}$ represents the $Vv$ of a target MODIS pixel, *i.e.*, the central MODIS pixel of the sliding window. Each equation corresponds to a neighbouring MODIS pixel in the 3 × 3 window. $P_{i,x,y}$ signifies the areal proportion of the *i*th land cover type within this MODIS pixel, indicating its area-weighted contribution. $V_{v,i,x,y}$ denotes the $Vv$ for the *i*th land cover type, which is assumed to be constant across the sliding window for each land cover type $i$, and is an unknown to be estimated.

By solving these equations, the $Vv$ value was inferred for each land cover type. The derived values of $V_{v,i,x,y}$ were then assigned as the $Vv$ for all the 30 m pixels of land cover type $i$ within the MODIS pixel at $(x, y)$. The same procedure was applied to obtain $V_{s,i,x,y}$. Figure 4 illustrates the method for downscaling 500 m pure NDVI values to 30 m resolution. ”

[Figure]

**Figure 4: The approach for downscaling the 500 m MultiVI *Vv* and *Vs* to 30 m resolution based on the GlobeLand30 product.”**

**R2C3:** I have two significant concerns about the data used to validate/assess their products. The first is the rather unsophisticated way the authors have applied the 'statistical' method. They have only used 3 years of data to derive statistics about $Vv$ and $Vs$. What if that period was continually wet, or continually dry, or was fire affected? The derived statistics cannot be assumed to be representative of that site. The authors have the ability to use a much longer time series and should do so. Also, the authors have applied the method with the expectation that it will work everywhere, which it is known not to. The method cannot return reliable $Vs$ values in heavily vegetated areas nor $Vv$ in sparsely vegetated areas. While the authors acknowledge this in the conclusion, this knowledge hasn't been applied in their design of the derivation of the statistically derived $Vs$ $Vv$ data. And so it is no surprise that this product performs poorly in these respective situations. This led the authors to conclude that (line 442)

"Traditional statistical methods are impractical to achieve this goal due to their reliance on pure pixels."

This is not universally true. More sophisticated implementations of the statistical method can be quite effective. Can the authors at least provide some more context to the reader about the simplicity of their approach relative to alternative approaches? Or maybe the authors could restrict the application of their statistical method to where it is known to be valid and hence avoid reporting values where it quite rightly doesn't work. None of this will change the excellent result that the multi-VI method is superior.

Re: **Done.** Thank you for your thorough and constructive comments. We have carefully revised the manuscript to address your concerns regarding the statistical methods used to derive *Vv* and *Vs*.

1) To improve the robustness of the statistical endmembers, we recalculated *Vv* and *Vs* pixel by pixel using a longer time series of Landsat data. Specifically, the maximum and minimum NDVI values from the period 2010–2020 were used to represent *Vv* and *Vs*, respectively. The corresponding description has been updated in Section 2.2.1 and Section 3.3.1.

*P6: Line 143~145*

The time-series Landsat 8 SR images from 2010 to 2020 were analyzed to derive statistical *Vv* and *Vs*. Pixels identified as cloud, cloud shadow, water, and snow in the Landsat 8 images were excluded using the corresponding quality assessment data.

*P13: Line 281~282*

The statistical method utilized Landsat 8 data from 2010 to 2020, processed on the Google Earth Engine (GEE) platform. The pixel-wise maximum and minimum NDVI values from 2010 to 2020 were set as the *Vv* and *Vs*, respectively.

2) Following this update, the rationality of the statistical endmembers has improved: *Vv* values increased in humid regions, while *Vs* values decreased in arid areas (Figure 5). Furthermore, we revised all the validation results using the updated statistical *Vv* and *Vs*, demonstrating improved accuracy compared to the values calculated using three years of data (for full details please refer to the revised Section 4).

3) We clarified that the statistical method used in this study yields reasonable FVC estimation in most regions, except in evergreen forest areas and extremely arid zones. The updated content demonstrates that, outside of these extreme regions, the statistical endmembers provide reliable FVC (Figure 9). The practicability of the statistical method has now been explicitly stated, and the corresponding analysis has been supplemented in Sections 4.2 and 4.3.

4) We fully acknowledge the practicality of statistical methods, especially their simplicity and reasonable accuracy in a suitable area. In fact, the MultiVI model proposed in this study incorporates statistical endmembers as boundaries for inversion. We have revised the manuscript in the introduction and discussion sections to avoid overgeneralized or dismissive statements regarding statistical methods, and we have expanded our discussion to better reflect their strengths and appropriate use cases. Additionally, we have cited the recent work by Donohue and

Renzullo (2025), which demonstrates that improved statistical implementations, when used with appropriate constraints, can yield reliable results. For full details please refer to Section5 (P20: Line 413~423) and R2C5.

**R2C4:** The second concern I have about the data used to validate/assess their products relates to how the field data at Heihe were derived. In scaling the field observations from 10 x 10 m to 90 x 90 m, the authors have effectively turned the field observations into a modelled product with its own errors. I would expect that a direct comparison between the 10 m field data and the 30 m *Vs Vv* data would provide a more robust comparison than upscaling the field data.

Re: **Done.** Thank you for your insightful comment. Following your suggestion, we have revised the validation approach for the Heihe site. Specifically, we directly compared the $10 \times 10$ m field-measured FVC with the 30 m MultiVI and statistical FVC, instead of upscaling the field data. The updated validation results are now presented in Figure 9, and corresponding revisions have been made in Section 4.3.

*P18: Line 388~390*

"The MultiVI FVC exhibits a slightly lower RMSD of 0.1285 compared to 0.1694 for the statistical FVC at the Heihe site. However, it has a lower $R^2$ of 0.7712 versus 0.8033 for the statistical FVC. The majority of the data points depicted in Figures 9c and 9d are positioned above the 1:1 line, indicating an overestimation in both the MultiVI FVC and the statistical FVC."

[Figure]

Figure 9: Scatterplots of the MultiVI FVC and the statistical FVC versus the field-measured FVC. The R² and RMSD values are also shown. N is the number of samples.

**R2C5:** One last comment is that some recent work is of direct relevance to this Mutil-VI paper (Donohue and Renzullo, 2025; https://doi.org/10.1071/BT24060). I expect this would have been published after the current manuscript's submission; however, it may be of interest to the authors. In making this statement I should also disclose that this is my paper (it's Randall Donohue here).

Re: **Done.** Thank you for sharing your recent work and for disclosing your authorship. We appreciate the valuable contribution of your study, which proposes improvements to traditional statistical methods and demonstrates their effectiveness in estimating FVC over Australia. We have cited this reference in the revised manuscript to acknowledge that the statistical method adopted in our study is relatively simple, while more

advanced implementations, such as yours, can achieve a higher level of estimation accuracy.

Newly added Reference:

Donohue, R. J. and Renzullo, L. J.: An assessment of the accuracy of satellite-derived woody and grass foliage cover estimates for Australia, Australian Journal of Botany, 73, https://doi.org/10.1071/BT24060, 2025.

*P20: Line 429~439*

In this study, statistical maps of *Vv* and *Vs* were generated using a common statistical criterion (Section 3.3.1). Empirical NDVI for fully vegetated pixels is typically reported to exceed 0.5 in most studies (Gao et al., 2020; Zeng et al., 2000; Montandon and Small, 2008), whereas the statistical *Vv* are observed to be below 0.3 in arid regions of China's mainland (Figure 5a). Except for these extreme areas, the statistical method demonstrates comparable accuracy to the MultiVI algorithm (Figures 7, 8, and 9c, 9d). The simple statistical methods face challenges in identifying appropriate NDVI values when pure pixels are lacking. In contrast, several studies have adopted more sophisticated statistical approaches to produce accurate FVC estimates across various ecosystems (Donohue and Renzullo, 2025; Donohue et al., 2018). These enhanced statistical methods take into account the inherent characteristics of vegetation in different ecosystems, employing separate thresholds to determine endmembers in arid and vegetated areas, respectively. The use of spatially adaptive thresholding facilitates the acquisition of reasonable endmembers and achieves high FVC accuracy, with a RMSE of approximately 0.1 (Donohue and Renzullo, 2025).

**R2C6:** Lines 42 and 49. The VI-based mixture model referred by the authors is specifically the NDVI-based mixture model. It is not a generic model that can use any vegetation index.

Re: **Done.** Thank you for your helpful comment. While the current study focuses on the NDVI-based mixture model, we would like to clarify that both the VI-based mixture model and the proposed MultiVI method are applicable to other vegetation indices, such as the Enhanced Vegetation Index (EVI). Previous studies that applied MultiVI to EVI have also reported high accuracy (Song et al., 2022a). To prevent any confusion, we have added a sentence in the introduction to clarify that the model framework is not limited to NDVI.

*P2: Line 62~64*

"The 'greenness' VIs that exhibit a strong linear relationship with FVC can be used in the VI-based mixture model, such as the enhanced vegetation index (EVI) and the normalized difference vegetation index (NDVI) (Song et al., 2022a)."

**R2C7:** Line 173. Doesn't look like the UAV data were used for anything at the Hebei site. Do they need to be mentioned at all?

Re: **Done.** Thank you for pointing this out. At the Hebei site, the grassland and cropland data were indeed acquired using UAV, and these data were included in the validation

analysis. We have revised the text to clarify the role of the UAV data and avoid potential confusion.

*P9: Line 182~184*

"For the cropland and grassland, an unmanned aerial vehicle (UAV) was used to capture FVC images, achieving a resolution higher than 1.5 cm and covering a plot size of 100 m × 100 m. The UAV-derived FVC values were used in the subsequent validation analysis."

**R2C8:** Line 200. It is a misconception that the NDVI has a saturation effect. When compared to foliage cover (which it what is has been shown to be linearly related to) there is no 'saturation'. This misconception arises when NDVI is incorrectly expected to bear some relationship with leaf area.

Re: **Done.** Thank you for your suggestion. We have removed the reference to the "saturation effect" in the revised text to avoid this common misconception.

Although NDVI and FVC exhibit an approximately linear relationship under the assumption of pure pixels, several studies have demonstrated that, in practice, NDVI often shows a nonlinear response to FVC in mixed pixels due to various factors (Mu et al., 2024). This nonlinear relationship can be attributed to soil background variability, sub-pixel shadow fractions, remote sensing geometry, terrain effects, and particularly the scale effect (Mu et al., 2024). Several studies have employed a nonlinear VI-based mixture model to estimate FVC, achieving greater accuracy than the linear model (Montandon and Small, 2008). We have revised the sentence in the manuscript to prevent any potential misunderstandings.

Newly added Reference:

Mu, X., Yang, Y., Xu, H., Guo, Y., Lai, Y., McVicar, T. R., Xie, D., and Yan, G.: Improvement of NDVI mixture model for fractional vegetation cover estimation with consideration of shaded vegetation and soil components, Remote Sensing of Environment, 314, 114409, https://doi.org/10.1016/j.rse.2024.114409, 2024.

*P9: Line 209~211*

"This coefficient accommodates the potential nonlinear relationship between FVC and NDVI, as shown in Eq. (3) (Xiao and Moody, 2005; Jiapaer et al., 2011; Choudhury et al., 1994; Mu et al., 2024):

$$F(\theta) = \left(\frac{V(\theta) - V_S}{V_v - V_S}\right)^k, \tag{3}"$$

**R2C9:** Line 229. Calling the values derived from a single year (2014) the 'historical' values is counterintuitive. They are not representative of site history.

Re: **Done.** Thank you for your comment. To avoid any misunderstanding, we have removed the term 'historical' and revised the sentence to clarify that the minimum and maximum NDVI values were derived from all available observations for the year 2014.

*P11: Line 237~239*

"The minimum and maximum NDVI values for each pixel in 2014 were used as statistical boundaries. This approach ensured that the derived *Vs* values remained below

the annual minimum, while the derived *Vv* values exceeded the annual maximum."

**R2C10:** Line 229. How much does using statistics derived from only one year of data (2014) limit the accuracy of the method when applied to other years? I would think it important to derive these 'historical' values from as long a time series as possible (which would be 23 or so years for MODIS).

Re: **Done.** Thank you for your comment. In response, we have revised the manuscript and addressed this issue from the following perspectives:

1) We acknowledge the importance of assessing the representativeness of single-year data. As detailed in our response to R1C1, we conducted a supplementary analysis comparing *Vv* and *Vs* from 2014 with those from 2018. The results indicate minimal interannual differences. Additionally, we clarified that the NDVI values of pure vegetation and bare soil pixels are generally stable across years, unless influenced by abrupt disturbances. For a comprehensive justification and supporting evidence, please refer to our response to R1C1.

2) The MultiVI algorithm estimates *Vv* and *Vs* by solving equations derived from two angular observations with significantly different NDVI values. As long as the differences between the angular observations are sufficient, a valid solution can be obtained. A single year of MODIS data captured a complete vegetation growth cycle, ensuring the availability of angular NDVI pairs with sufficient contrast for reliable inversion. Our experiments show that introducing too many angular observations can lead to overfitting, which degrades the estimation accuracy of *Vv* and *Vs*. Therefore, selecting a representative set of well-separated angular observations from one year is an effective strategy to ensure solution quality while avoiding overfitting.